# Sleep-promoting effects of threonine link amino acid metabolism in *Drosophila* neuron to GABAergic control of sleep drive

Yoonhee Ki, Chunghun Lim*

School of Life Sciences, Ulsan National Institute of Science and Technology, Ulsan, Republic of Korea

**Abstract** Emerging evidence indicates the role of amino acid metabolism in sleep regulation. Here we demonstrate sleep-promoting effects of dietary threonine (SPET) in *Drosophila*. Dietary threonine markedly increased daily sleep amount and decreased the latency to sleep onset in a dose-dependent manner. High levels of synaptic GABA or pharmacological activation of metabotropic GABA receptors (GABA$_B$-R) suppressed SPET. By contrast, synaptic blockade of GABAergic neurons or transgenic depletion of GABA$_B$-R in the ellipsoid body R2 neurons enhanced sleep drive non-additively with SPET. Dietary threonine reduced GABA levels, weakened metabotropic GABA responses in R2 neurons, and ameliorated memory deficits in plasticity mutants. Moreover, genetic elevation of neuronal threonine levels was sufficient for facilitating sleep onset. Taken together, these data define threonine as a physiologically relevant, sleep-promoting molecule that may intimately link neuronal metabolism of amino acids to GABAergic control of sleep drive via the neuronal substrate of sleep homeostasis.
**Editorial note:** This article has been through an editorial process in which the authors decide how to respond to the issues raised during peer review. The Reviewing Editor's assessment is that all the issues have been addressed (see decision letter).
DOI: https://doi.org/10.7554/eLife.40593.001

*For correspondence:
clim@unist.ac.kr

## Introduction

The circadian clock and sleep homeostasis are two key regulators that shape daily sleep behaviors in animals (*Borbély, 1982*). In stark contrast to the homeostatic nature of sleep, the internal machinery of sleep is vulnerable to external (e.g., environmental change) or internal conditions (e.g., genetic mutation) that lead to adaptive changes in sleep behaviors. Sleep behavior is conserved among mammals, insects, and even lower eukaryotes (*Allada and Siegel, 2008*; *Joiner, 2016*). Since the identification of the voltage-gated potassium channel *Shaker* as a sleep-regulatory gene in *Drosophila* (*Cirelli et al., 2005*), fruit flies have been one of the most advantageous genetic models to dissect molecular and neural components that are important for sleep homeostasis and plasticity.

To date, a number of sleep-regulatory genes and neurotransmitters have been identified in animal models as well as in humans (*Allada et al., 2017*; *Artiushin and Sehgal, 2017*; *Tomita et al., 2017*). For instance, the inhibitory neurotransmitter gamma-aminobutyric acid (GABA) is known to have a sleep-promoting role that is conserved in invertebrates and vertebrates. Hypomorphic mutations in mitochondrial *GABA-transaminase* (*GABA-T*) elevate GABA levels and lengthen baseline sleep in flies (*Chen et al., 2015*). The long sleep phenotype in *GABA-T* mutants accompanies higher sleep consolidation and shorter latency to sleep onset, consistent with the observations that pharmacological enhancement of GABAergic transmission facilitates sleep in flies and mammals, including

humans (*Holmes and Sugden, 1975*; *Lancel et al., 1998*; *Schneider et al., 1977*). In addition, *resistance to dieldrin* (*Rdl*), a *Drosophila* homolog of the ionotropic GABA receptor, suppresses wake-promoting circadian pacemaker neurons in adult flies to exert sleep-promoting effects (*Agosto et al., 2008*; *Chung et al., 2009*; *Liu et al., 2014a*; *Parisky et al., 2008*). Similarly, 4,5,6,7-tetrahydroisoxazolo[5,4 c]pyridin-3-ol (THIP), an agonist of the ionotropic GABA receptor, promotes sleep in insects and mammals (*Dissel et al., 2015*; *Faulhaber et al., 1997*; *Lancel, 1997*).

Many sleep medications modulate GABAergic transmission. A prominent side effect of anti-epileptic drugs relevant to GABA is causing drowsiness (*Jain and Glauser, 2014*). Conversely, glycine supplements improve sleep quality in a way distinct from traditional hypnotic drugs, minimizing deleterious cognitive problems or addiction (*Bannai and Kawai, 2012*; *Yamadera et al., 2007*). In fact, glycine or D-serine acts as a co-agonist of N-methyl-D-aspartate receptors (NMDARs) and promotes sleep through the sub-type of ionotropic glutamate receptors (*Dai et al., 2019*; *Kawai et al., 2015*; *Tomita et al., 2015*). Emerging evidence further supports the roles of amino acid transporters and metabolic enzymes in sleep regulation (*Aboudhiaf et al., 2018*; *Sonn et al., 2018*; *Stahl et al., 2018*). In particular, we have demonstrated that starvation induces the expression of metabolic enzymes for serine biosynthesis in *Drosophila* brains, and elevates free serine levels to suppress sleep via cholinergic signaling (*Sonn et al., 2018*). These observations prompted us to hypothesize that other amino acids may also display neuro-modulatory effects on sleep behaviors.

## Results

### Dietary threonine promotes sleep and facilitates sleep onset

To determine if amino acid supplements modulate sleep in *Drosophila*, we employed an infrared beam-based *Drosophila* activity monitor (DAM) that detects locomotor activity in individual flies (*Pfeiffenberger et al., 2010*). Sleep behaviors in wild-type flies fed 5% sucrose containing 17.5 mM of each amino acid were quantitatively assessed in 12 hr light:12 hr dark (LD) cycles at 25°C. The strongest impact on sleep quantity and quality was observed with cysteine supplementation (*Figure 1A* and *Figure 1—figure supplement 1*). However, dietary cysteine compromised locomotion and caused high lethality during our sleep assay (see *Figure 1—figure supplement 6*). We thus excluded it from further analyses. Intriguingly, threonine supplementation potently elevated total sleep amount by increasing the number of sleep bouts (*Figure 1A* and *Figure 1—figure supplement 1*). In addition, dietary threonine evidently shortened the latency to sleep onset after lights-off. The sleep-promoting effects of dietary threonine (SPET) were dose-dependent and observed in both male and female flies (*Figure 1—figure supplement 2*). Transgenic silencing of sensory neurons that express either gustatory receptors (*Gr66a*, *Gr33a*, and *Gr5a*) or olfactory co-receptor (*Lone et al., 2016*) negligibly affected SPET as compared to relevant heterozygous controls (*Figure 1—figure supplement 3*). These results suggest that sensory perception of dietary threonine is less likely responsible for SPET. We further found that flies fed nutrient-rich food containing additional protein sources (e.g., cornmeal, yeast) also exhibited SPET, although higher concentrations of threonine were required (*Figure 1—figure supplement 4*). We reason that flies may ingest smaller volume of daily food on nutrient-rich diet than on sucrose-only diet as a compensation for their difference in calories per volume (*Carvalho et al., 2005*). Nonetheless, these data indicate that SPET is not limited to carbohydrate-only diets.

It has previously been shown that flies exhibit a positional preference relative to their food source, depending on sleep-wake cycles or genetic backgrounds (*Donelson et al., 2012*). These observations raised the possibility that threonine supplementation might have affected the positional preference in wild-type flies, thereby leading to the overestimation of their sleep amount by the DAM-based analyses (*Figure 1—figure supplement 5A*). To exclude this possibility, we placed individual flies into circular arenas in which food is provided unilaterally from the whole floor (*Figure 1—figure supplement 5B and C*). Locomotor activities of individual flies were then video-recorded in LD cycles. The video-based assessment of sleep behaviors in control- versus threonine-fed flies further confirmed SPET (*Figure 1—figure supplement 5D*). Lower waking activity (i.e., beam crosses per minute during wakefulness) was observed in threonine-fed flies by the DAM analysis (*Figure 1—figure supplement 1*). Dietary threonine actually shortened total traveling distance, but it did not significantly affect moving speed in the video analysis (*Figure 1—figure supplement 5D* and

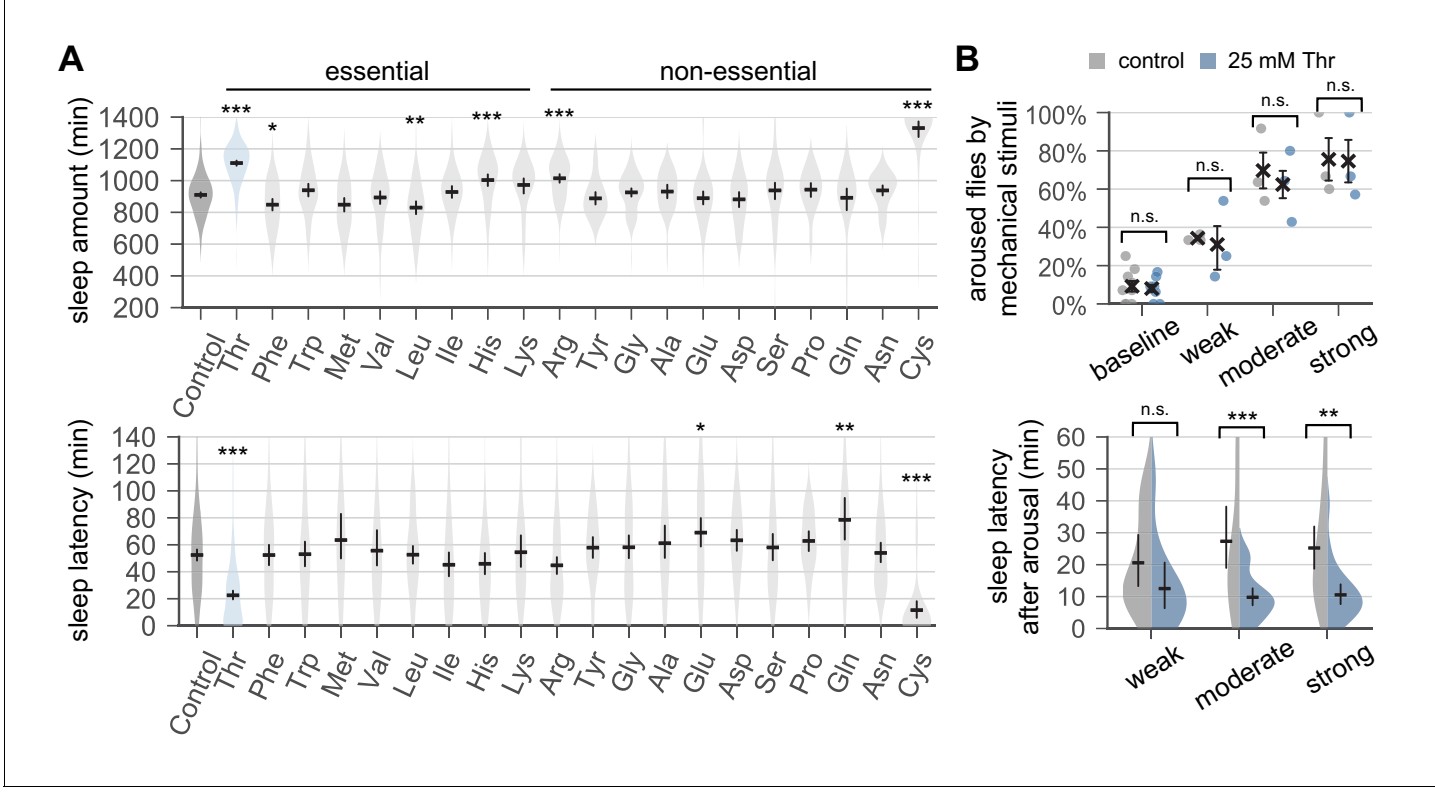

**Figure 1.** Dietary threonine promotes sleep and facilitates sleep onset. (**A**) Wild-type male flies were individually loaded on to 5% sucrose food containing 17.5 mM of each amino acid (day 0) and entrained in LD cycles at 25°C. Total sleep amount (top) and latency to sleep onset after lights-off (bottom) were calculated in individual flies on day 4 and averaged for each amino acid. Essential and non-essential amino acids are grouped separately as shown at the top. The width of a violin plot indicates the density of samples. The violins are restricted by the observed ranges. Error bars indicate mean ±95% confidence interval (CI) (n = 29–213). *p<0.05, **p<0.01, ***p<0.001 to control (i.e., no amino acid supplement) as determined by one-way ANOVA, Dunnett's multiple comparisons test. (**B**) Control- and threonine-fed flies were awakened by a range of mechanical stimuli 4 hr after lights-off on day 4. Aroused flies were defined if they displayed no activity for >5 min prior to the stimulus but showed any locomotor response within 10 min. The percentage of aroused flies per condition was averaged from three independent experiments (top). Error bars indicate mean ± SEM (n = 3). Sleep latency after arousal was calculated in individual flies and averaged for each condition (bottom). Error bars indicate mean ±95% CI (n = 12–27). Two-way ANOVA detected significant effects of dietary threonine on sleep latency after arousal (F[1,119]=20.43, p<0.0001) but not on % aroused flies (F[1,16] =0.227, p=0.6402). n.s., not significant; **p<0.01, ***p<0.001 as determined by Bonferroni's multiple comparisons test.

DOI: https://doi.org/10.7554/eLife.40593.002

The following figure supplements are available for figure 1:

**Figure supplement 1.** Dietary threonine increases the number of sleep bouts but decreases waking activities.

DOI: https://doi.org/10.7554/eLife.40593.003

**Figure supplement 2.** Sleep-promoting effects of dietary threonine (SPET) are dose-dependent and observed in both male and female flies.

DOI: https://doi.org/10.7554/eLife.40593.004

**Figure supplement 3.** Transgenic silencing of sensory neurons that express either gustatory or olfactory receptors does not abolish SPET.

DOI: https://doi.org/10.7554/eLife.40593.005

**Figure supplement 4.** Wild-type flies fed protein-rich food display SPET comparably to those fed sucrose-based food.

DOI: https://doi.org/10.7554/eLife.40593.006

**Figure supplement 5.** A video-based sleep analysis validates SPET in threonine-fed flies.

DOI: https://doi.org/10.7554/eLife.40593.007

**Figure supplement 6.** Dietary threonine does not impair general locomotion.

DOI: https://doi.org/10.7554/eLife.40593.008

**Figure supplement 7.** Dietary threonine induces a higher sleep drive.

DOI: https://doi.org/10.7554/eLife.40593.009

**Figure supplement 8.** Genetic loss of *Lk* or *Lkr* function does not affect SPET.

DOI: https://doi.org/10.7554/eLife.40593.010

*Figure 1—figure supplement 6*). Therefore, it is unlikely that threonine supplementation causes general locomotor impairment responsible for low waking activity or long sleep phenotypes. It is also noteworthy that low waking activity does not necessarily associate with long sleep phenotypes as observed with tryptophan supplementation (*Figure 1A* and *Figure 1—figure supplement 1*).

To examine if SPET affects arousal threshold (i.e., sleep depth), we quantified arousal responses to sensory stimuli during sleep. Control- and threonine-fed flies displayed no significant differences in the percentage of flies aroused by a given range of mechanical stimuli in the middle of night (*Figure 1B*). However, threonine-fed flies displayed shorter latency to the first post-stimulus bout of sleep. Consistent results were obtained when nighttime sleep was interrupted by a pulse of light (*Figure 1—figure supplement 7*). Taken together, these data suggest that a higher sleep drive, but not a change in sleep depth, may contribute to SPET.

## Circadian clock-dependent control of sleep onset is dispensable for SPET

*Rdl* and *wide awake* (*wake*) are two evolutionarily conserved genes that contribute to circadian clock-dependent control of sleep onset in *Drosophila* (*Agosto et al., 2008*; *Liu et al., 2014a*). A circadian transcription factor, CLOCK (CLK), drives daily rhythmic transcription of *wake*, particularly in a subset of clock neurons that express the circadian neuropeptide PIGMENT-DISPERSING FACTOR (PDF) (*Liu et al., 2014a*). Subsequently, WAKE acts as a clock output molecule that interacts with RDL, silences the wake-promoting PDF neurons, and facilitates sleep onset. Therefore, we asked whether circadian clocks and their regulation of sleep drive would be necessary for SPET.

We first confirmed that female mutants trans-heterozygous for hypomorphic *Rdl* alleles had shorter sleep latency in control-fed condition than their heterozygous controls (*Agosto et al., 2008*) (*Figure 2—figure supplement 1*). Dietary threonine, however, shortened sleep latency additively with the loss of *Rdl* function (p=0.084, by two-way ANOVA). In addition, trans-heterozygous *Rdl* mutation did not compromise SPET on daily sleep amount compared to either heterozygous controls. We next examined if SPET was suppressed in arrhythmic clock mutants. Loss of *Clk* function caused long sleep latency in fed condition (*Liu et al., 2014a*), and SPET had additive effects on the latency phenotype in *Clk* mutants (*Figure 2A*, p=0.14 by two-way ANOVA). On the other hand, short sleep latency in *per* mutants (*Liu et al., 2014a*) likely caused a floor effect, leading to no significant SPET on their sleep latency (*Figure 2A*). Nonetheless, wild-type and both clock mutants showed comparable SPET on daily sleep amount (*Figure 2A*, p=0.8367 for *Clk* mutants; p=0.2573 for *per* mutants by two-way ANOVA). Finally, it has been shown that overexpression of dominant-negative CLK proteins (CLK$^{DN}$) in PDF neurons is sufficient to abolish free-running circadian locomotor rhythms (*Tanoue et al., 2004*) and lengthen sleep latency (*Liu et al., 2014a*). We observed consistent effects of CLK$^{DN}$ overexpression in PDF neurons on sleep drive in control-fed condition, but it did not suppress SPET (*Figure 2B*). These lines of our genetic evidence suggest that SPET does not require clock-dependent control of sleep onset by circadian clock genes or PDF neurons.

To further test the implication of circadian clocks in SPET, we compared SPET in different light-dark conditions. Constant dark (DD) following LD entrainment eliminates masking behaviors in direct response to the light transitions while allowing free-running circadian rhythms by endogenous clocks (*Allada and Chung, 2010*). We found that DD did not suppress SPET but rather exaggerated it particularly in male flies (*Figure 2C and D*, p<0.0001 to SPET on sleep amount or sleep latency in LD by two-way ANOVA). SPET was thus evident even in the absence of light. By contrast, constant light (LL) abolishes circadian rhythms in wild-type flies (*Emery et al., 2000*). Consequently, control-fed flies completely lost their daily rhythms in sleep-wake cycles (*Figure 2C and D*) and dampened their sleep latency in LL (*Figure 2—figure supplement 2*). This caused a floor effect whereby SPET was barely detectable, particularly on sleep latency at the transition of subjective day and night in LL as compared to LD. Nonetheless, we observed significant effects of dietary threonine on sleep latency (i.e., shorter sleep latency in threonine-fed flies) when SPET on sleep latency was compared among different time-points in LL (p=0.0003 to control-fed male in LL; p<0.0001 to control-fed female in LL by two-way ANOVA). Dietary threonine also increased daily sleep amount in LL (*Figure 2C and D*). In fact, male flies displayed comparable SPET on daily sleep amount in LD and LL (p=0.1835 by two-way ANOVA). Collectively, these data support that higher sleep drive by SPET likely operates in a manner independent of circadian clocks and their control of sleep onset.

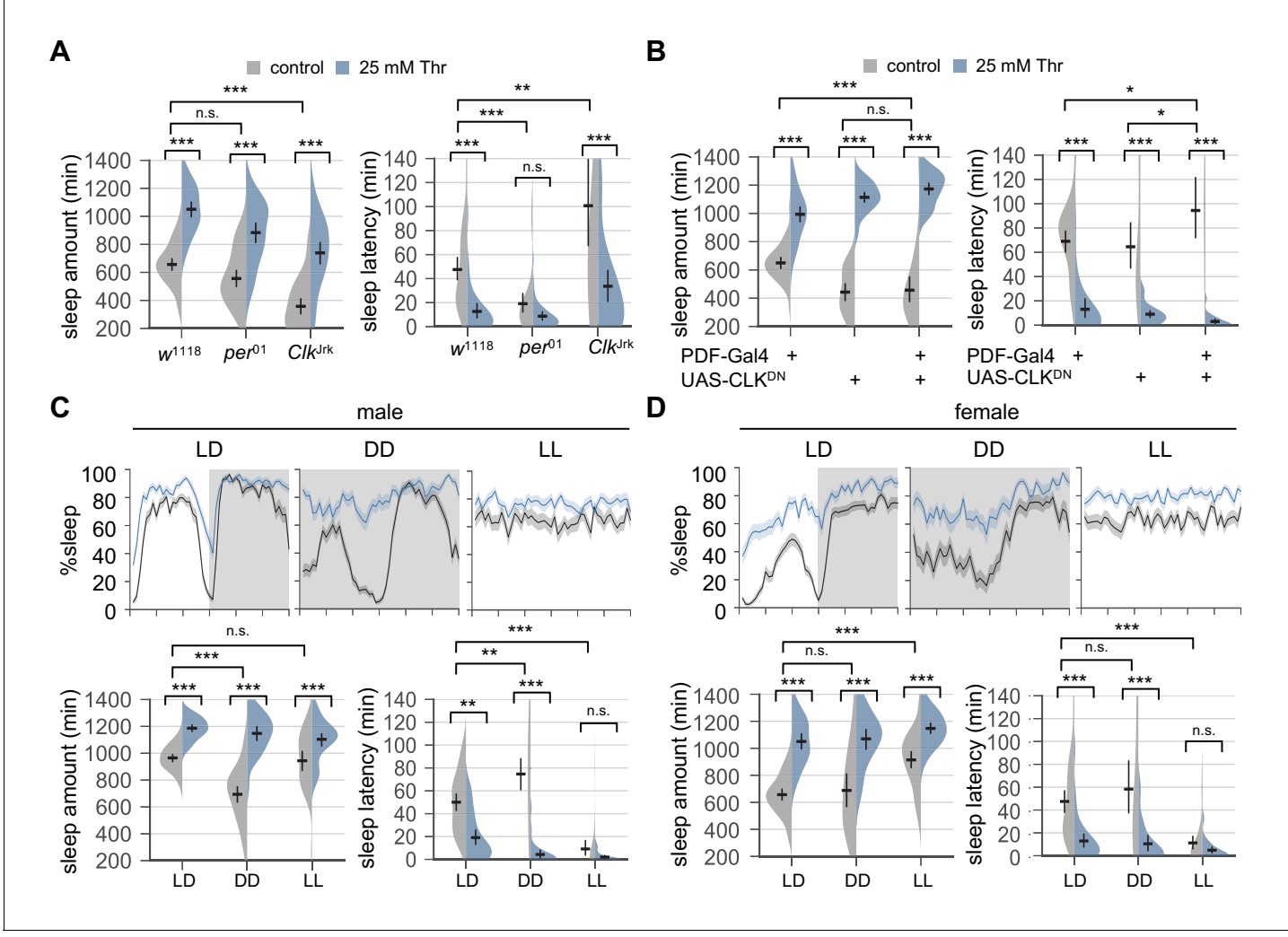

**Figure 2.** Circadian rhythms and clock-dependent control of sleep onset are dispensable for SPET. (**A**) Arrhythmic clock mutants were loaded on to 5% sucrose food containing the indicated amount of threonine (day 0) and entrained in LD cycles at 25°C. Sleep behaviors in individual female flies were analyzed similarly to the data presented in *Figure 1A*. Two-way ANOVA detected no significant interaction of SPET with *per*[01] (F[1,159]=1.293, p=0.2573 for sleep amount) or *Clk*[Jrk] (F[1,160]=0.0426, p=0.8367 for sleep amount; F[1,160]=2.199, p=0.14 for sleep latency). Error bars indicate mean ±95% CI (n = 35–46). (**B**) PDF neuron-specific overexpression of dominant-negative CLK proteins (CLK[DN]) lengthened sleep latency in female flies fed control food (5% sucrose) but it did not suppress SPET. Error bars indicate mean ±95% CI (n = 26–42). (**C and D**) Wild-type flies were loaded on to 5% sucrose food containing the indicated amount of threonine (day 0) and then entrained in LD or constant light (LL) cycles at 25°C. For sleep analyses in constant dark (DD), LD-entrained flies were transferred to DD at the end of day 4 and their sleep was monitored during the first DD cycle (day 5). Averaged sleep profiles (% sleep per 30 min bin) on day 4 (LD or LL) or day 5 (DD) were shown at the top. Data represent mean ± SEM (n = 25–46). Error bars in the violin plots indicate mean ±95% CI (n = 25–46). n.s., not significant; *p<0.05, **p<0.01, ***p<0.001 as determined by two-way ANOVA, Tukey's multiple comparisons test.

DOI: https://doi.org/10.7554/eLife.40593.011

The following figure supplements are available for figure 2:

**Figure supplement 1.** Genetic loss of *Rdl* function does not suppress SPET.
DOI: https://doi.org/10.7554/eLife.40593.012

**Figure supplement 2.** Constant light (LL) strongly dampens daily rhythms in sleep-wake cycles and sleep latency, but SPET is detectable in LL.
DOI: https://doi.org/10.7554/eLife.40593.013

## Genetic or pharmacological elevation of synaptic GABA suppresses SPET

To elucidate genetic and neural mechanisms underlying SPET, we examined effects of dietary threonine on sleep behaviors in loss-of-function mutants of other sleep-regulatory genes. Interestingly,

SPET was potently suppressed in *GABA-T* mutants trans-heterozygous for a null allele (*GABA-T*[PL]) over chromosomal deficiency (*Figure 3—figure supplement 1*). Their sensitivity to SPET was partially but significantly rescued by transgenic overexpression of wild-type GABA-T (*Chen et al., 2015*). However, the trans-heterozygosity of these strong *GABA-T* alleles promoted sleep in control-fed condition (*Chen et al., 2015*), raising the possibility that a ceiling effect may mask SPET. We thus tested SPET in weaker allelic combinations of *GABA-T* mutations. *GABA-T* mutants trans-heterozygous for null over hypomorphic alleles (*GABA-T*[F] or *GABA-T*[LL]) (*Chen et al., 2015*) did not significantly affect baseline sleep in control-fed condition as compared to their heterozygous controls (*Figure 3A*). Nonetheless, these mutants still exhibited the resistance to SPET (p<0.0001 to SPET on sleep amount or sleep latency in heterozygous controls by two-way ANOVA).

To independently confirm the implication of *GABA-T* function in SPET, we pharmacologically silenced the enzymatic activity of GABA-T in wild-type flies by oral administration of ethanolamine O-sulfate (EOS), a GABA-T inhibitor. EOS supplement did not significantly increase daily sleep amount at a given dose in our sleep assay, but it modestly shortened sleep latency in wild-type flies fed control food (*Figure 3B*). However, SPET was suppressed in EOS-fed flies (p<0.0001 to SPET on sleep amount or sleep latency in control flies by two-way ANOVA) similarly as in *GABA-T* mutants. Considering that GABA-T is a mitochondrial enzyme which metabolizes GABA into succinic semialdehyde (*Chen et al., 2015*), we hypothesized that high GABA levels at GABAergic synapses might interfere with sleep drive by dietary threonine. This idea was further supported by our observation that nipecotic acid (NipA), which blocks GABA reuptake from synaptic clefts (*Leal and Neckameyer, 2002*), comparably suppressed SPET (*Figure 3B*, p<0.0001 to SPET on sleep amount or sleep latency in control flies by two-way ANOVA). Collectively, these genetic and pharmacological data suggest that SPET may involve a sleep drive relevant to GABA. In addition, evidence from our adult-specific manipulations of GABA levels excludes possible developmental effects of *GABA-T* mutation or GABA on SPET.

## Dietary threonine decreases GABA and glutamate levels

Genetic deficit in the metabolic conversion of GABA to glutamate leads to high levels of GABA in *GABA-T* mutants while they have low levels of glutamate and alpha-ketoglutarate, a glutamate derivative that enters tricarboxylic cycle (*Maguire et al., 2015*). These changes in GABA-derived metabolites are accompanied with impairment in energy homeostasis as supported by the high ratio of NAD[+]/NADH levels and low ATP levels in *GABA-T* mutants. Consequently, *GABA-T* mutants cannot survive on carbohydrate-based food (i.e., 5% sucrose +1.5% agar) but their metabolic stress phenotypes are rescued by the supplement of glutamate and other amino acids that can be metabolized to glutamate. We thus asked if dietary threonine would induce relevant metabolic changes that may be responsible for SPET.

Dietary threonine did not significantly affect ATP levels or the ratio of NAD[+]/NADH levels (*Figure 4—figure supplement 1A*). However, pyruvate levels were selectively elevated in threonine-fed flies (p<0.0001 to succinate by one-way ANOVA, Dunnett's multiple comparisons test), possibly due to the metabolism of dietary threonine into pyruvate via L-2-amino-acetoacetate (*Figure 4—figure supplement 2*). Nonetheless, dietary pyruvate itself did not promote sleep (*Figure 4—figure supplement 1B*). Quantification of free amino acids further revealed that dietary threonine reduced the relative levels of proline, histidine, alanine, glutamate, and GABA among other amino acids (*Figure 4A*, p<0.05 or p<0.001 to arginine by one-way ANOVA, Dunnett's multiple comparisons test). Since it has been shown that glutamate acts as either a wake- or sleep-promoting neurotransmitter in *Drosophila* (*Guo et al., 2016*; *Robinson et al., 2016*; *Tomita et al., 2015*; *Zimmerman et al., 2017*), we asked if co-administration of threonine and glutamate could suppress SPET. Glutamate supplement, however, negligibly affected SPET (*Figure 4—figure supplement 3*, p=0.91 for sleep amount; p=0.516 for sleep latency by two-way ANOVA), suggesting that dietary threonine may not limit glutamate levels to promote sleep. It is noteworthy that glutamate supplement can rescue metabolic stress, but not sleep phenotypes, in *GABA-T* mutants, indicating independent regulatory pathways of GABA-relevant metabolism and sleep (*Maguire et al., 2015*).

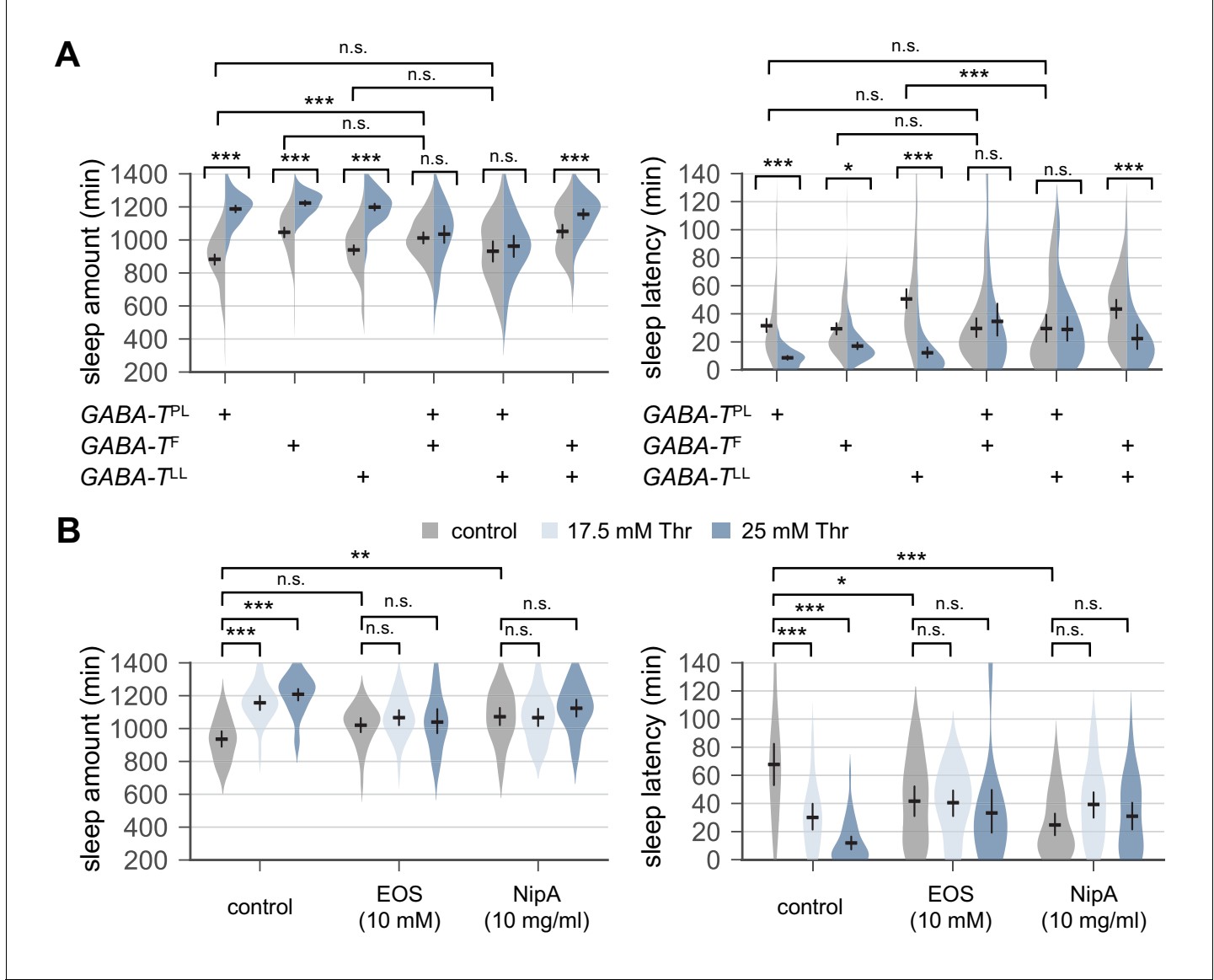

**Figure 3.** Genetic or pharmacological elevation of GABA suppresses SPET. (**A**) *GABA-T* trans-heterozygous mutants were resistant to SPET. Sleep behaviors in individual male flies were analyzed similarly to the data presented in *Figure 1A*. Two-way ANOVA detected significant suppression of SPET in *GABA-T* trans-heterozygous mutants on sleep amount (F[2,403]=39.21, p<0.0001 for *GABA-T^PL^/GABA-T^F^*; F[2,430]=32.28, p<0.0001 for *GABA-T^PL^/GABA-T^LL^*; F[2,454]=13.99, p<0.0001 for *GABA-T^F^/GABA-T^LL^*) and sleep latency (F[2,403]=13.35, p<0.0001 for *GABA-T^PL^/GABA-T^F^*; F[2,430]=15.97, p<0.0001 for *GABA-T^PL^/GABA-T^LL^*; F[2,454]=9.324, p=0.0001 for *GABA-T^F^/GABA-T^LL^*), compared to their heterozygous controls. Error bars indicate mean ±95% CI (n = 32–114).(**B**) Co-administration of GABA-T inhibitor (EOS) or GABA transporter inhibitor (NipA) with threonine blocked SPET in wild-type flies. Where indicated, EOS or NipA was added to the sucrose food containing the increasing amounts of threonine. Sleep behaviors were analyzed as described above. Two-way ANOVA detected significant interaction of SPET with EOS (F[2,155]=14.07, p<0.0001 for sleep amount; F[2,155]=11.2, p<0.0001 for sleep latency) or NipA (F[2,162]=13.09, p<0.0001 for sleep amount; F[2,162]=26.58, p<0.0001 for sleep latency). Error bars indicate mean ±95% CI (n = 22–37) .n.s., not significant; *p<0.05, **p<0.01, ***p<0.001 as determined by Tukey's multiple comparisons test.

DOI: https://doi.org/10.7554/eLife.40593.014

The following figure supplement is available for figure 3:

**Figure supplement 1.** Transgenic overexpression of wild-type GABA-T partially rescues baseline sleep and SPET in *GABA-T* mutants.

DOI: https://doi.org/10.7554/eLife.40593.015

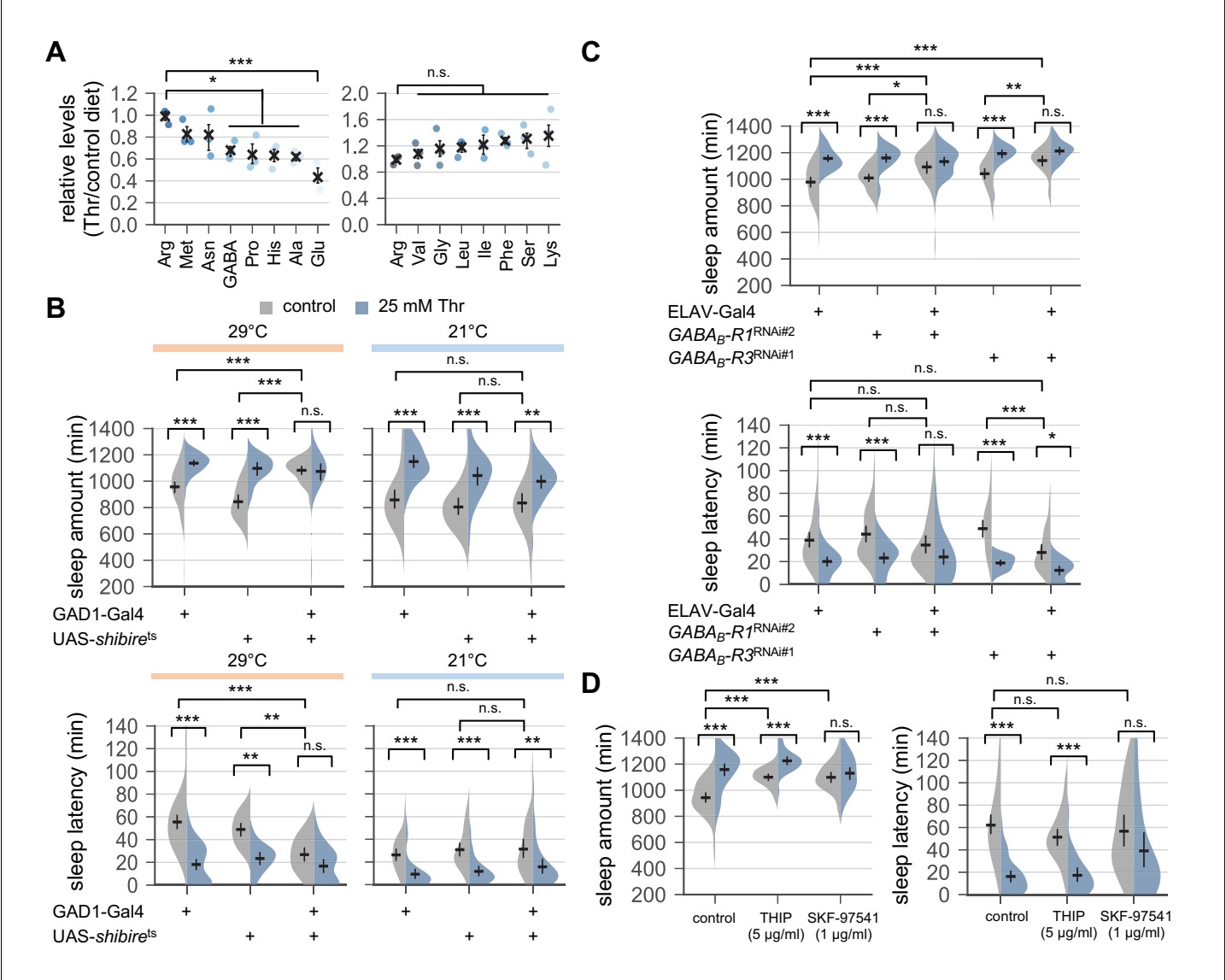

**Figure 4.** Down-regulation of metabotropic GABA transmission likely mediates SPET. (**A**) Dietary threonine decreased the relative levels of select amino acids including GABA and glutamate. Wild-type male flies were loaded on to standard cornmeal-yeast-agar food containing either 0 mM (control) or 50 mM threonine, and then entrained in LD cycles at 25°C for 4 days before harvest. Relative levels of free amino acids in head extracts from threonine-fed flies were measured using ion exchange chromatography and then normalized to those in head extracts from control-fed flies. Error bars indicate mean ± SEM (n = 3). n.s., not significant; *p<0.05, ***p<0.001 to the relative levels of arginine as determined by one-way ANOVA, Dunnett's multiple comparisons test. (**B**) Conditional blockade of GABAergic transmission promoted sleep in control-fed condition and masked SPET. Transgenic flies expressing a temperature-sensitive allele of *shibire* (*shibire*ts) were loaded on to 5% sucrose food containing the indicated amount of threonine (day 0) and entrained in LD cycles at 29°C (restrictive) or 21°C (permissive). Sleep behaviors in individual male flies were analyzed on day 3 (29°C) or day 6 (21°C) since low temperature delayed SPET even in heterozygous controls. Two-way ANOVA detected significant masking of SPET by *shibire*ts overexpression in GAD1-expressing cells at 29°C (F[2,197]=14.06, p<0.0001 for sleep amount; F[2,196]=6.953, p=0.0012 for sleep latency), but not at 21°C (F[2,184] =2.055, p=0.131 for sleep amount; F[2,184]=0.1835, p=0.8325 for sleep latency) as compared to their heterozygous controls. Error bars indicate mean ±95% CI (n = 14–63 for 29°C; n = 25–35 for 21°C). (**C**) Pan-neuronal deletion of metabotropic GABA receptors (GABAB-R1 and GABAB-R3) by transgenic RNA interference (RNAi) increased daily sleep amount in control-fed condition and masked SPET. Locomotor activities in individual male flies were monitored similarly to the data presented in *Figure 1A*. Sleep behaviors were analyzed on day 3 to better compare the sensitivity to SPET among different genotypes. Two-way ANOVA detected significant masking of SPET by the pan-neuronal RNAi on sleep amount (F[2,236]=8.913, p=0.0002 for *GABAB-R1*RNAi#2; F[2,317]=16.78; F[2,193]=4.594, p=0.0112 for *GABAB-R3*RNAi#1) and sleep latency (F[2,193]=3.267, p=0.0403 for *GABAB-R3*RNAi#1) as compared to their heterozygous controls. Error bars indicate mean ±95% CI (n = 24–50). (**D**) Oral administration of SKF-97541 (an agonist of metabotropic GABA receptor), but not of THIP (an agonist of ionotropic GABA receptor), suppressed SPET. Sleep behaviors in individual male flies were analyzed as described above. Where indicated, THIP (5 μg/ml) or SKF-97541 (1 μg/ml) was added to the behavior food. Two-way ANOVA

*Figure 4 continued on next page*

*Figure 4 continued*

detected significant effects of SKF-97541 on SPET (F[1,143]=17.39, p<0.0001 for sleep amount; F[1,143]=6.898, p=0.0096 for sleep latency).Error bars indicate mean ±95% CI (n = 27–53). n.s., not significant; *p<0.05, **p<0.01, ***p<0.001 as determined by Tukey's multiple comparisons.

DOI: https://doi.org/10.7554/eLife.40593.016

The following figure supplements are available for figure 4:

**Figure supplement 1.** Dietary threonine selectively elevates pyruvate levels but dietary pyruvate itself does not promote sleep.

DOI: https://doi.org/10.7554/eLife.40593.017

**Figure supplement 2.** A metabolic pathway of serine, glycine, and threonine.

DOI: https://doi.org/10.7554/eLife.40593.018

**Figure supplement 3.** Glutamate supplement does not suppress SPET.

DOI: https://doi.org/10.7554/eLife.40593.019

**Figure supplement 4.** Dietary threonine elevates intracellular $Ca^{2+}$ levels in a subset of GABAergic neurons.

DOI: https://doi.org/10.7554/eLife.40593.020

**Figure supplement 5.** Pan-neuronal depletion of metabotropic GABA receptor R1, but not R2, affects SPET.

DOI: https://doi.org/10.7554/eLife.40593.021

**Figure supplement 6.** Structural and functional relevance of alpha-ketobutyric acid, a threonine derivative, to GABA and GABA derivatives.

DOI: https://doi.org/10.7554/eLife.40593.022

## Down-regulation of GABA transmission via metabotropic GABA receptors induces sleep and masks SPET

To determine if dietary threonine affects GABA transmission, we examined intracellular $Ca^{2+}$ levels in glutamate decarboxylase 1 (GAD1)-expressing GABAergic neurons as a quantitative proxy for their neural activity. Since threonine supplementation exhibited cumulative effects on baseline sleep in LD cycles, we reasoned that it might be necessary to monitor the long-term changes in neural activity associated with threonine diet. Accordingly, we employed a transgenic reporter of the calcium-sensitive transcriptional activator LexA (CaLexA) that translocates into nucleus in a calcium-dependent manner and induces the expression of green fluorescent protein (GFP) (*Masuyama et al., 2012*). Confocal microscopy of adult fly brains revealed the strongest GFP expression by the GABAergic CaLexA in neurons projecting into antennal lobe (AL), medial antenno-cerebral tract (mACT), and lateral horn (LH) among other GAD1-expressing neurons (*Figure 4—figure supplement 4A*). These observations suggest a heterogeneity in baseline $Ca^{2+}$ levels among GABAergic neuron subsets. Interestingly, threonine, but not arginine, induced the CaLexA signal in a subset of GABAergic neurons adjacent to the antennal lobe (LN, lateral neurons) (*Figure 4—figure supplement 4B and C*). By contrast, no detectable changes were observed in the CaLexA signals from other sleep-regulatory loci such as mushroom body or dopaminergic neurons upon threonine diet (*Figure 4—figure supplement 4D and E*). Although the sensitivity of CaLexA may limit the detectable size and duration of $Ca^{2+}$ changes in our experimental condition, these results support the relative specificity of $Ca^{2+}$ response in LN to the threonine diet. Given that dietary threonine decreased GABA levels, GABAergic LN may selectively display a compensatory increase in their neural activity. Alternatively, it is possible that auto-inhibitory GABA receptors (*Pinard et al., 2010*) are expressed more strongly in these LN than other GABAergic neurons. Low GABA levels in threonine-fed flies may then relieve this negative feedback and stimulate their neural activity upon threonine diet. In either case, these results prompted us to ask if GABAergic transmission would be necessary for SPET.

To further validate the implication of GABAergic transmission in SPET, we expressed a *shibire*[ts] transgene (*Kitamoto, 2001*) in GAD1-expressing GABAergic neurons. The *shibire*[ts] is a temperature-sensitive mutant allele in a *Drosophila* homolog of dynamin that interferes with synaptic vesicle recycling and thus, blocks synaptic transmission at restrictive (29°C) but not permissive (21°C) temperature. The conditional blockade of synaptic transmission in GABAergic neurons induced sleep in control-fed condition (*Figure 4B*), and it significantly masked SPET (p<0.0001 for sleep amount; p=0.0012 for sleep latency by two-way ANOVA). These long sleep phenotypes were partially but consistently observed by the pan-neuronal depletion of metabotropic GABA receptor R1 or R3 ($GABA_B$-R1 or $GABA_B$-R3) (*Figure 4C*). However, their effects were in contrast with those observed by hypomorphic *GABA-T* mutations that suppressed SPET but did not promote baseline sleep in control-fed condition. We further found that co-administration of an agonist of metabotropic GABA

receptors (SKF-97541), but not of ionotropic GABA receptors (THIP), with threonine suppressed SPET particularly on sleep latency (*Figure 4D*, p=0.1285 for THIP; p=0.0096 for SKF-97541 by two-way ANOVA). Adult-specific manipulations of GABAergic transmission by the temperature-sensitive allele or by the oral administration of receptor-specific agonists excluded possible developmental effects of GABA on SPET. Collectively, these data suggest a possible model that SPET involves the down-regulation of metabotropic GABA transmission to induce sleep whereas genetic or pharmacological elevation of the GABA transmission interferes with this process to suppress SPET. Nonetheless, the multimeric nature of GABA receptors and their expression in either wake- or sleep-promoting neurons likely complicate the net effects of general activation or silencing of GABA transmission on sleep. We thus asked if more specific suppression of the metabotropic GABA transmission in a dedicated neural locus would induce sleep and mask SPET, thereby supporting our hypothesis above.

## Metabotropic GABA transmission in ellipsoid body R2 neurons contributes to SPET

A previous study mapped a subset of ellipsoid body (EB) neurons in the adult fly brain (hereafter referred to as R2 EB neurons) as a neural locus important for sleep homeostasis (*Liu et al., 2016*). Neural activity in R2 EB neurons positively correlates to sleep need, and the transgenic excitation of R2 EB neurons is sufficient to induce rebound sleep. Considering that SPET involves a higher sleep drive, we hypothesized that dietary threonine might affect the activity of R2 EB neurons via metabotropic GABA transmission. Since the intracellular signaling downstream of metabotropic GABA receptors triggers cAMP synthesis (*Onali et al., 2003*), we employed Epac1-camps, a transgenic fluorescence resonance energy transfer (FRET) sensor for cyclic adenosine monophosphate (cAMP) (*Shafer et al., 2008*) (*Figure 5A*). Our live-brain imaging of the Epac1-camps in R2 EB neurons detected a dose-dependent increase in cAMP levels by a bath application of GABA (*Figure 5—figure supplement 1A*). Pre-incubation with tetrodotoxin did not affect the GABA-induced elevation of cAMP levels, indicating cell-autonomous effects of GABA on these R2 EB neurons (*Figure 5—figure supplement 1B*). We further found that dietary threonine modestly, but significantly, dampened the GABA response in R2 EB neurons (*Figure 5B*), validating that dietary threonine modulates the neural activity of this homeostatic sleep driver.

We next asked if metabotropic GABA transmission in R2 EB neurons would contribute to SPET. The RNAi-mediated depletion of GABA$_B$-R2 or GABA$_B$-R3 in R2 EB neurons modestly promoted sleep in control-fed conditions (*Figure 5C*). Moreover, it significantly masked SPET on sleep amount (p=0.0093 for GABA$_B$-R2; p=0.0007 for GABA$_B$-R3 by two-way ANOVA) and on sleep latency (p=0.0072 for GABA$_B$-R2; p<0.0001 for GABA$_B$-R3 by two-way ANOVA), as compared to heterozygous controls. The GABA$_B$-R3 RNAi phenotypes were consistent with those observed by the pan-neuronal depletion of GABA$_B$-R3 (*Figure 4C*). On the other hand, no detectable phenotypes were observed by the pan-neuronal overexpression of the GABA$_B$-R2 RNAi transgenes, likely due to insufficient depletion of GABA$_B$-R2 in R2 EB neurons by the pan-neuronal driver (*Figure 4—figure supplement 5*). Nonetheless, these results indicate that genetic suppression of the metabotropic GABA transmission in R2 EB neurons phenotypically mimics SPET at the levels of neural activity (i.e., weaker GABA responses) and sleep behaviors (i.e., higher sleep drive). The sleep phenotypes by the pan-neuronal, but not R2 EB-specific, depletion of GABA$_B$-R1 (*Figure 4C* and *Figure 5—figure supplement 2*) further suggest that this sub-type of metabotropic GABA receptors may be expressed in non-R2 EB neurons to mediate sleep-regulatory transmission relevant to SPET.

## SPET rescues short-term memory in fly mutants with memory deficit

Inhibitory effects of dietary threonine on metabotropic GABA transmission in R2 EB neurons support that SPET enhances sleep drive via a physiologically relevant neural locus. Nonetheless, the operational definition of a sleep episode in our behavioral assays (i.e., no movement for longer than 5 min) could mislead threonine-induced behavioral quiescence into SPET. Therefore, we took two independent approaches to validate that SPET is physiologically relevant to sleep. Sleep deprivation impairs learning in *Drosophila* (*Seugnet et al., 2008*). By contrast, genetic or pharmacological induction of sleep ameliorates memory deficits in plasticity mutants (*Dissel et al., 2015*). These observations have convincingly demonstrated the physiological benefits of sleep in memory formation, and we

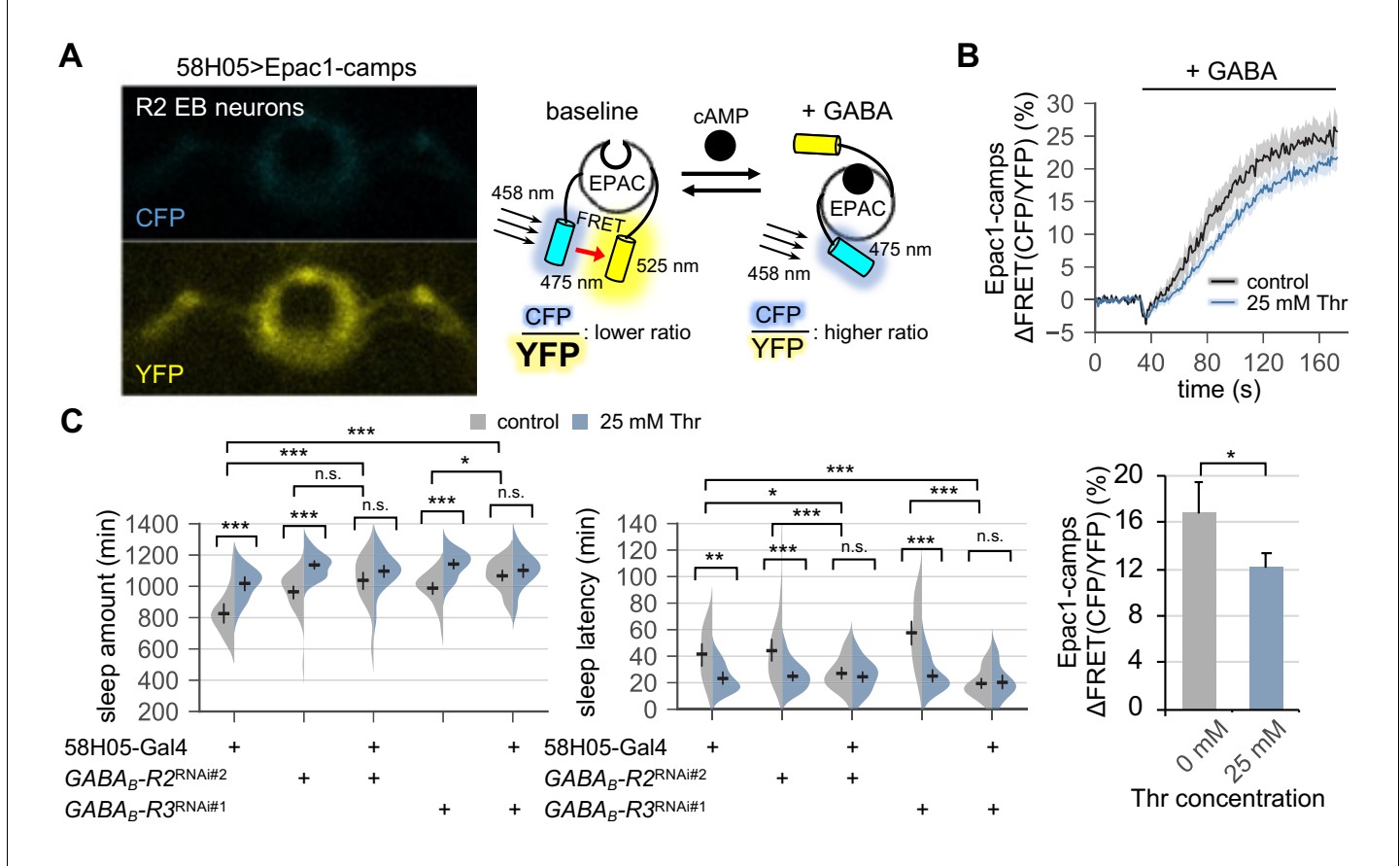

**Figure 5.** Metabotropic GABA transmission in ellipsoid body R2 neurons contributes to SPET. (**A**) A representative live-brain image of Epac1-camps (a transgenic FRET sensor for cAMP) expressed in R2 EB neurons by 58H05-Gal4 driver (left). An inverse correlation between intracellular cAMP levels and FRET intensity was depicted on the right. CFP, cyan fluorescent protein; YFP, yellow fluorescent protein. (**B**) Transgenic flies (58H05 > Epac1 camps) were fed on control or threonine-containing food for 4 days in LD cycles at 25°C. Whole brains were dissected out and transferred to an imaging chamber. A time series of the fluorescence images was recorded using a multi-photon microscopy. Where indicated, 100 mM GABA was batch-applied to the imaging medium. FRET analysis was performed using ZEN software. Averaged histograms of the relative changes in FRET intensity (top) and their averaged median values (bottom) were shown. Data represent mean ± SEM (n = 10–14). *p<0.05 as determined by Student's *t*-test. (**C**) The RNAi-mediated deletion of metabotropic GABA receptors (GABA_B-R2 and GABA_B-R3) in R2 EB neurons induced sleep in control-fed condition and masked SPET. Sleep behaviors in individual male flies were monitored similarly to the data presented in *Figure 4C*. Two-way ANOVA detected significant masking of SPET by the overexpression of RNAi transgenes in R2 EB neurons on sleep amount (F[2,161]=4.818, p=0.0093 for GABA_B-R2$^{RNAi\#2}$; F[2,133]=7.669, p=0.0007 for GABA_B-R3$^{RNAi\#1}$) and sleep latency (F[2,161]=5.088, p=0.0072 for GABA_B-R2$^{RNAi\#2}$; F[2,133]=14.65, p<0.0001 for GABA_B-R3$^{RNAi\#1}$) as compared to their heterozygous controls. Error bars indicate mean ±95% CI (n = 17–34). n.s., not significant; *p<0.05, **p<0.01, ***p<0.001 as determined by Tukey's multiple comparisons.

DOI: https://doi.org/10.7554/eLife.40593.023

The following figure supplements are available for figure 5:

**Figure supplement 1.** R2 EB neurons are GABA-ceptive.

DOI: https://doi.org/10.7554/eLife.40593.024

**Figure supplement 2.** Transgenic depletion of metabotropic GABA receptor R1 in R2 EB neurons does not affect SPET.

DOI: https://doi.org/10.7554/eLife.40593.025

thus hypothesized that dietary threonine should rescue memory mutants if it would induce physiologically relevant sleep. To test this hypothesis, we employed a short-term memory (STM) test that was based on aversive phototaxic suppression (*Seugnet et al., 2009*) (*Figure 6A*), and examined possible effects of dietary threonine on STM.

Hypomorphic mutants of D1-like dopamine receptor 1 (*dumb²*) displayed impairment in STM (*Figure 6B*), consistent with previous observation (*Seugnet et al., 2008*). Dietary threonine substantially improved STM in *dumb* mutants (*Figure 6B*), and comparably rescued memory deficit in

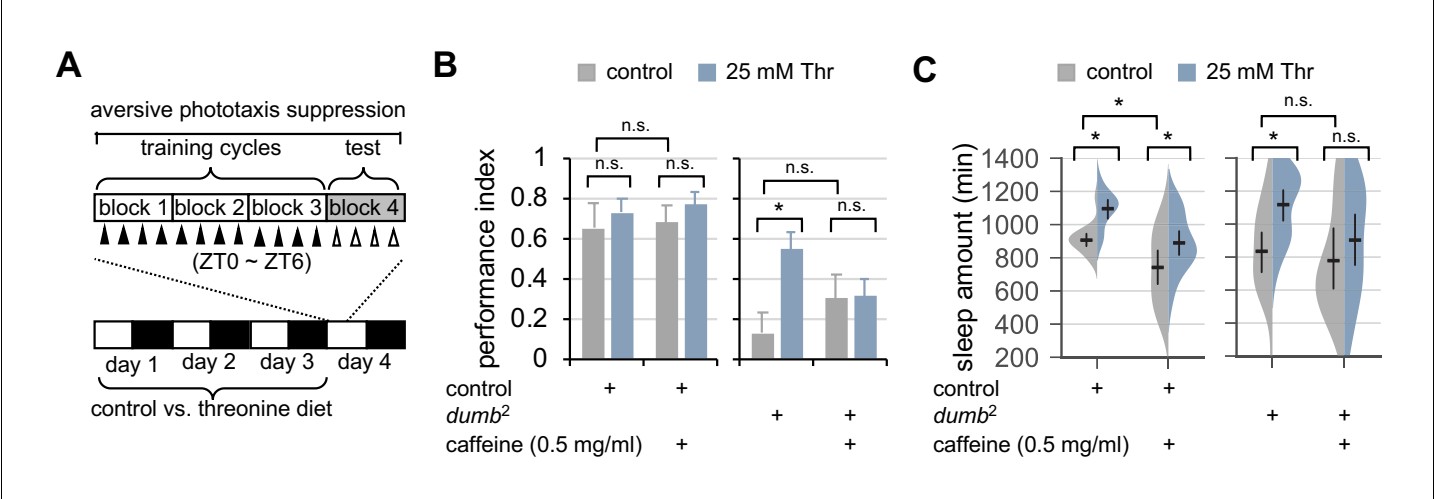

**Figure 6.** Dietary threonine rescues short-term memory in *dumb* mutants with memory deficit in a sleep-dependent manner. (**A**) An experimental design of the short-term memory (STM) test after three cycles of training on aversive phototaxis suppression. Wild-type (Canton S) or *dumb²* mutant flies were individually loaded on to 5% sucrose food containing either 0 mM (control) or 25 mM threonine (day 0), and then entrained for 3 days in LD cycles at 25˚C. Where indicated, 0.5 mg/ml of caffeine was added to the behavior food. Locomotor activity in individual male flies was monitored using the DAM system to analyze sleep behaviors on day 3 prior to the STM test on day 4. (**B**) The performance index during the test session was calculated in individual flies and averaged for each condition. Two-way ANOVA detected no significant effects of threonine or caffeine on STM in control flies (F [1,41]=0.9644, p=0.3318 for threonine; [1,41]=0.1433, p=0.7070 for caffeine). By contrast, two-way ANOVA detected significant interaction between threonine and caffeine on STM in *dumb* mutants (F[1,43]=4.329, p=0.0435). Data represent average ± SEM (n = 10–13). (**C**) Sleep behaviors in individual male flies were analyzed similarly to the data presented in *Figure 1A*. Two-way ANOVA detected significant effects of threonine or caffeine on daily sleep amount in control flies (F[1,62]=18.41, p<0.0001 for threonine; F[1,62]=22.26, p<0.0001 for caffeine), but not their significant interaction (F[1,62] =0.2836, p=0.5963). Additive effects of threonine and caffeine on daily sleep amount were also observed in *dumb* mutants (F[1,56]=1.091, p=0.3007 by two-way ANOVA). Error bars indicate mean ±95% CI (n = 11–19). n.s., not significant; *p<0.05 as determined by Tukey's multiple comparisons test.
DOI: https://doi.org/10.7554/eLife.40593.026

The following figure supplement is available for figure 6:

**Figure supplement 1.** Dietary threonine rescues short-term memory in *rutabaga* mutants with memory deficit.
DOI: https://doi.org/10.7554/eLife.40593.027

plasticity mutants of *rutabaga*, a *Drosophila* homolog of adenylate cyclase (*Dissel et al., 2015*) (*Figure 6—figure supplement 1*). To confirm that memory rescue actually requires threonine-induced sleep, we pharmacologically deprived sleep in *dumb* mutants by oral administration of caffeine (*Andretic et al., 2008*; *Nall et al., 2016*; *Wu et al., 2009*), and tested its effects on the threonine-dependent rescue of STM in *dumb* mutants. SPET and caffeine-induced arousal displayed additive effects on daily sleep amount in control flies (*Figure 6C*, p=0.5963 by two-way ANOVA) while negligibly affecting their performance index in the memory test (*Figure 6B*). Consistent with the implication of dopaminergic activation in caffeine-induced arousal (*Andretic et al., 2008*; *Nall et al., 2016*), baseline sleep in *dumb* mutants were relatively insensitive to caffeine. Co-administration of caffeine and threonine, however, suppressed *dumb* mutant sleep more evidently than caffeine alone (*Figure 6B*), and blocked the improvement of their memory deficit by dietary threonine (*Figure 6C*, p=0.0435 by two-way ANOVA).

## Genetic elevation of endogenous threonine levels facilitates sleep onset

We next asked if a physiologically relevant increase in threonine levels could act as an endogenous promoter of sleep. We hypothesized that genetic mutations in threonine-metabolizing enzymes might elevate the steady-state levels of endogenous threonine. *CG5955* is a fly homolog of threonine 3-dehydrogenase that converts threonine and NAD⁺ into L-2-amino-acetoacetate, NADH, and H⁺ (*Figure 7A*). We identified a transposable P-element insertion in the proximal promoter region of the *CG5955* locus that reduced the relative levels of *CG5955* mRNA (*Figure 7B and C*). Biochemical

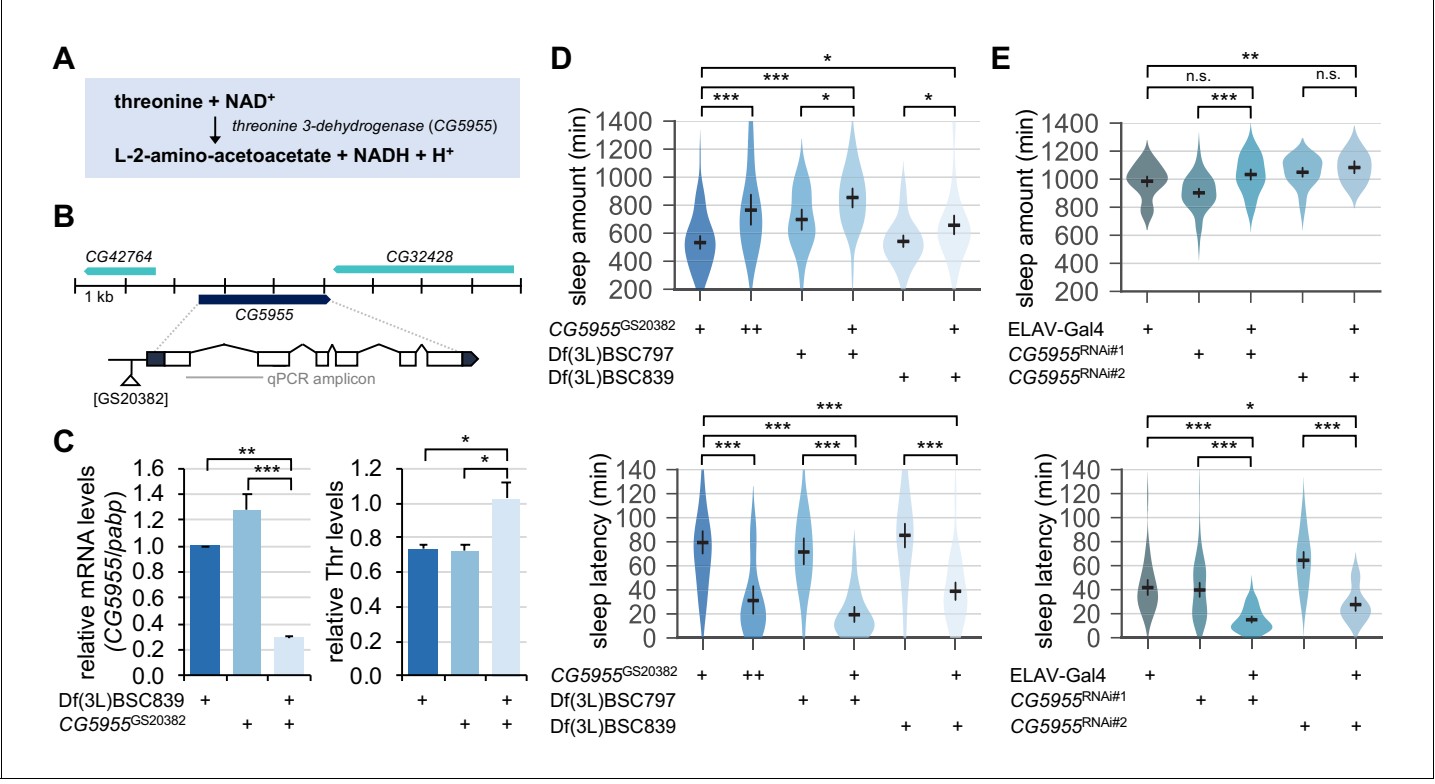

**Figure 7.** Genetic suppression of threonine 3-dehydrogenase elevates endogenous threonine levels and facilitates sleep onset. (**A**) A threonine metabolism catalyzed by threonine 3-dehydrogenase (*CG5955*). (**B**) A hypomorphic mutant allele of the P element insertion ([GS20382]) in the *CG5955* locus. An amplicon used in quantitative PCR was depicted by a gray line. (**C**) Trans-heterozygous mutations in *CG5955* reduced *CG5955* mRNA levels (left, normalized to *polyA-binding protein* mRNA levels) but elevated endogenous threonine levels (right, normalized to protein levels). Data represent mean ± SEM (n = 3). *p<0.05, **p<0.01, ***p<0.001 as determined by one-way ANOVA, Tukey's multiple comparisons test. (**D**) Loss-of-function mutations in *CG5955* promoted sleep. *CG5955* mutants were loaded on to 5% sucrose (day 0) and entrained in LD cycles at 25°C. Sleep behaviors in individual female flies were analyzed on day 3 and averaged for each genotype. Error bars indicate mean ±95% CI (n = 26–76). *p<0.05, ***p<0.001 as determined by one-way ANOVA, Tukey's multiple comparisons test. (**E**) Pan-neuronal depletion of *CG5955* expression shortened sleep latency. Sleep behaviors in individual male flies were analyzed as described above due to the X-chromosomal insertion of the pan-neuronal ELAV-Gal4 driver. DICER-2 was co-expressed with each of two independent RNAi transgenes (CG5955^RNAi #1 and CG5955^RNAi #2) to enhance the RNAi effects. Error bars indicate mean ±95% CI (n = 20–64). n.s., not significant; *p<0.05, **p<0.01, ***p<0.001 as determined by one-way ANOVA, Dunnett's multiple comparisons test.
DOI: https://doi.org/10.7554/eLife.40593.028

analyses of fly extracts confirmed that *CG5955* mutants trans-heterozygous for the hypomorphic allele over chromosomal deficiency displayed a higher ratio of threonine to total protein levels than heterozygous controls (*Figure 7C*). Behavioral analyses revealed that either homozygous or trans-heterozygous mutation in *CG5955* increased daily sleep amounts compared to heterozygous controls (*Figure 7D*). Moreover, the latency to sleep onset after lights-off was strongly shortened in *CG5955* mutants, indicating a high sleep drive. We further found that pan-neuronal depletion of *CG5955* expression was sufficient to mimic *CG5955* mutants in terms of their sleep latency phenotype (*Figure 7E*). Since genetic manipulations of a metabolic enzyme can lead to compensating changes in relevant metabolic pathways or development, we do not exclude the possibility that these indirect effects may have contributed to the higher sleep drive observed in *CG5955* mutants. Nonetheless, our genetic, biochemical, and behavioral evidence supports that threonine metabolism in the brain modulates sleep drive in flies.

## Discussion

The molecular and neural machinery of sleep regulation intimately interacts with external (e.g., light, temperature) and internal sleep cues (e.g., sleep pressure, metabolic state) to adjust the sleep

architecture in animals. Using a *Drosophila* genetic model, we have investigated whether dietary amino acids could affect sleep behaviors and thereby discovered SPET. Previous studies have demonstrated that the wake-promoting circadian pacemaker neurons are crucial for timing sleep onset after lights-off in LD cycles (*Agosto et al., 2008*; *Chung et al., 2009*; *Liu et al., 2014a*; *Parisky et al., 2008*). In addition, WAKE-dependent silencing of clock neurons and its collaborative function with RDL have been suggested as a key mechanism in the circadian control of sleep onset (*Liu et al., 2014a*). However, our evidence indicates that SPET facilitates sleep onset in a manner independent of circadian clocks. We further elucidate that SPET operates likely via the down-regulation of metabotropic GABA transmission in R2 EB neurons, a neural locus for generating homeostatic sleep drive (*Liu et al., 2016*).

Both food availability and nutritional quality substantially affect sleep behaviors in *Drosophila*. Sucrose contents in food and their gustatory perception dominate over dietary protein to affect daily sleep (*Catterson et al., 2010*; *Linford et al., 2012*; *Linford et al., 2015*). Starvation promotes arousal in a manner dependent on the circadian clock genes *Clock* and *cycle* (*Keene et al., 2010*) as well as neuropeptide F (NPF), which is a fly ortholog of mammalian neuropeptide Y (*Chung et al., 2017*). On the other hand, protein is one of the nutrients that contribute to the postprandial sleep drive in *Drosophila* (*Murphy et al., 2016*) and this observation is possibly relevant to SPET. While *Leucokinin* (*Lk*) and *Lk* receptor (*Lkr*) play important roles in dietary protein-induced postprandial sleep (*Murphy et al., 2016*) and in starvation-induced arousal (*Murakami et al., 2016*), we observed comparable SPET between hypomorphic mutants of *Lk* or *Lkr* and their heterozygous controls (*Figure 1—figure supplement 8*). Therefore, SPET and its neural basis reveal a sleep-regulatory mechanism distinct from those involved in sleep plasticity relevant to food intake.

What will be the molecular basis of SPET? Given the general implication of GABA in sleep promotion, a simple model will be that a molecular sensor expressed in a subset of GABAergic neurons (i.e., LN) directly responds to an increase in threonine levels, activates GABA transmission, and thereby induces sleep. Several lines of our evidence, however, favored the other model that dietary threonine actually down-regulates metabotropic GABA transmission in R2 EB neurons, de-represses the neural locus for generating homeostatic sleep drive, and thereby enhances sleep drive. The latter model does not necessarily conflict with sleep-promoting effects of genetic or pharmacological conditions that generally elevate GABA levels or enhance GABAergic transmission since those effects will be the net outcome of activated GABA transmission via various sub-types of GABA receptors expressed in either wake- or sleep-promoting neurons and their circuitry.

The structural homology among threonine, GABA, and their metabolic derivatives (e.g., alpha-ketobutyrate and gamma-hydroxybutyrate) led us to the hypothesis that these relevant chemicals may act as competitive substrates in enzymatic reactions for their overlapping metabolism (*Figure 4—figure supplement 6*). Consequently, dietary threonine may limit the total flux of GABA-glutamate-glutamine cycle possibly through substrate competition, decreases the size of available GABA pool, and thereby down-scales GABA transmission for SPET. This accounts for why genetic or pharmacological elevation of GABA levels rather suppresses SPET. Threonine, GABA, and their derivatives may also act as competitive ligands for metabotropic GABA receptors, explaining weak GABA responses in R2 EB neurons of threonine-fed flies. Biochemical and neural evidence supportive of this hypothesis is quite abundant. It has been previously shown that alpha-ketobutyrate, GABA, and the ketone body beta-hydroxybutyrate act as competitive substrates in common enzymatic reactions (*Beyerinck and Brass, 1987*; *Lund et al., 2011*; *Suzuki et al., 2009*). Moreover, functional interactions of beta-hydroxybutyrate or gamma-hydroxybutyrate with GABAergic signaling have been well documented (*Absalom et al., 2012*; *Carter et al., 2009*; *Carter et al., 2005*; *Lund et al., 2011*; *Nasrallah et al., 2010*; *Snead and Gibson, 2005*; *Suzuki et al., 2009*). Finally, threonine and GABA derivatives have anti-convulsive effects (*Growdon et al., 1991*; *Hauser et al., 1992*; *Lee et al., 1990*), which further support their common structural and functional relevance to GABAergic signaling.

The removal of the amino group is the initial step for amino acid metabolism, and various transaminases mediate its transfer between amino acids and alpha-keto acids. On the other hand, a group of amino acids (i.e., glutamate, glycine, serine, and threonine) has their own deaminases that can selectively remove the amino group (*Bender, 2014*). The presence of these specific deaminases is indicative of active mechanisms that individually fine-tune the baseline levels of these amino acids in metabolism, and possibly in the context of other physiological processes as well. This idea is further

supported by the conserved roles of glutamate, glycine, and serine as neurotransmitters or neuro-modulators important for brain function, including sleep regulation (*Kawai et al., 2015*; *Tomita et al., 2017*; *Zimmerman et al., 2017*). In fact, serine, glycine, and threonine constitute a common metabolic pathway (*Figure 4—figure supplement 2*), and threonine may contribute indirectly to glycine- or serine-dependent activation of sleep-promoting NMDAR (*Kawai et al., 2015*; *Tomita et al., 2015*). Nonetheless, we found that sleep-modulatory effects of dietary glycine were distinct from SPET and thus, we speculate that threonine may act as an independent neuromodulator, similar to other amino acids with their dedicated deaminases.

While several lines of our data support that threonine is likely to be an endogenous sleep driver in fed conditions, we have recently demonstrated that starvation induces serine biosynthesis in the brain and neuronal serine subsequently suppresses sleep via cholinergic signaling (*Sonn et al., 2018*). These two pieces of our relevant works establish a compelling model that the metabolic pathway of serine-glycine-threonine functions as a key sleep-regulatory module in response to metabolic sleep cues (e.g., food ingredients and dietary stress). We further hypothesize that the adaptive control of sleep behaviors by select amino acids and their conserved metabolic pathway suggests an ancestral nature of their sleep regulation. Future studies should address if the serine-glycine-threonine metabolic pathway constitutes the sleep homeostat that can sense and respond to different types of sleep needs. In addition, it will be interesting to determine if this metabolic regulation of sleep is conserved among other animals, including humans.

# Materials and methods

## Key resources table

| Reagent type (species) or resource | Designation | Source or reference | Identifiers | Additional information |
|---|---|---|---|---|
| Genetic reagent (*D. melanogaster*) | $w^{1118}$ | Bloomington Drosophila Stock Center | RRID:BDSC_5905 | |
| Genetic reagent (*D. melanogaster*) | Canton S | Korea Drosophila Resource Center | | Stock #K211 |
| Genetic reagent (*D. melanogaster*) | $CG5955^{GS20382}$ | Kyoto Drosophila Genomics and Genetics Resources | RRID:DGGR_201409 | |
| Genetic reagent (*D. melanogaster*) | Df(3L)BSC797 | Bloomington Drosophila Stock Center | RRID:BDSC_27369 | CG5955 deficiency |
| Genetic reagent (*D. melanogaster*) | Df(3L)BSC839 | Bloomington Drosophila Stock Center | RRID:BDSC_27917 | CG5955 deficiency |
| Genetic reagent (*D. melanogaster*) | $rut^{2080}$ | Bloomington Drosophila Stock Center | RRID:BDSC_9405 | |
| Genetic reagent (*D. melanogaster*) | $DA1^{dumb2}$ | Harvard Medical School | RRID: FlyBase_FBst1017920 | $Dop1R1^{f02676}$ |
| Genetic reagent (*D. melanogaster*) | ELAV-Gal4 | Bloomington Drosophila Stock Center | RRID:BDSC_458 | |
| Genetic reagent (*D. melanogaster*) | GAD1-Gal4 | Bloomington Drosophila Stock Center | RRID:BDSC_51630 | |
| Genetic reagent (*D. melanogaster*) | 58H05-Gal4 | Bloomington Drosophila Stock Center | RRID:BDSC_39198 | |

*Continued on next page*

*Continued*

| Reagent type (species) or resource | Designation | Source or reference | Identifiers | Additional information |
|---|---|---|---|---|
| Genetic reagent (*D. melanogaster*) | Gr5a-Gal4 | Bloomington Drosophila Stock Center | RRID:BDSC_57591 | |
| Genetic reagent (*D. melanogaster*) | Gr33a-Gal4 | Bloomington Drosophila Stock Center | RRID:BDSC_31425 | |
| Genetic reagent (*D. melanogaster*) | Gr66a-Gal4 | Bloomington Drosophila Stock Center | RRID:BDSC_28801 | |
| Genetic reagent (*D. melanogaster*) | Orco-Gal4 | Bloomington Drosophila Stock Center | RRID:BDSC_26818 | |
| Genetic reagent (*D. melanogaster*) | $Lk^{c275}$ | Bloomington Drosophila Stock Center | RRID:BDSC_16324 | |
| Genetic reagent (*D. melanogaster*) | Df(3L)Exel6123 | Bloomington Drosophila Stock Center | RRID:BDSC_7602 | *Lk* deficiency |
| Genetic reagent (*D. melanogaster*) | $Lkr^{c003}$ | Bloomington Drosophila Stock Center | RRID:BDSC_16250 | |
| Genetic reagent (*D. melanogaster*) | Df(3L)BSC557 | Bloomington Drosophila Stock Center | RRID:BDSC_25119 | *Lkr* deficiency |
| Genetic reagent (*D. melanogaster*) | $per^{01}$ | PMID: 9630223 | RRID:BDSC_80917 | |
| Genetic reagent (*D. melanogaster*) | $Clk^{Jrk}$ | PMID: 9630223 | RRID:BDSC_24515 | |
| Genetic reagent (*D. melanogaster*) | PDF-Gal4 | PMID: 10619432 | | |
| Genetic reagent (*D. melanogaster*) | $UAS-Clk^{DN}$ | *Tanoue et al., 2004* | RRID:BDSC_36318 | |
| Genetic reagent (*D. melanogaster*) | $Rdl^{MDRR}$ | Kyoto Drosophila Genomics and Genetics Resources | RRID:DGGR_106444 | |
| Genetic reagent (*D. melanogaster*) | $Rdl^{1}$ | Kyoto Drosophila Genomics and Genetics Resources | RRID:DGGR_106453 | |
| Genetic reagent (*D. melanogaster*) | $GABA-T^{PL}$ | Bloomington Drosophila Stock Center | RRID:BDSC_19461 | $GABAT^{PL00338}$, null mutants |
| Genetic reagent (*D. melanogaster*) | $GABA-T^{F}$ | Harvard Medical School | RRID: FlyBase_FBst101711 | $GABAT^{f01602}$, hypomorphic |
| Genetic reagent (*D. melanogaster*) | $GABA-T^{LL}$ | Kyoto Drosophila Genomics and Genetics Resources | RRID:DGGR_141269 | $GABAT^{LL04492}$, hypomorphic |
| Genetic reagent (*D. melanogaster*) | UAS-GABA-T | *Chen et al., 2015* | RRID:FlyBase_FBst0491743 | |
| Genetic reagent (*D. melanogaster*) | Df(3L)BSC731 | Bloomington Drosophila Stock Center | RRID:BDSC_26829 | *GABA-T* deficiency |

*Continued on next page*

*Continued*

| Reagent type (species) or resource | Designation | Source or reference | Identifiers | Additional information |
|---|---|---|---|---|
| Genetic reagent (*D. melanogaster*) | UAS-*shibire*[ts] | *Kitamoto, 2001* | | |
| Genetic reagent (*D. melanogaster*) | 30Y-Gal4 | Bloomington Drosophila Stock Center | RRID:BDSC_30818 | |
| Genetic reagent (*D. melanogaster*) | TH-Gal4 | Bloomington Drosophila Stock Center | RRID:BDSC_8848 | |
| Genetic reagent (*D. melanogaster*) | UAS-mLexA-VP16-NFAT | *Masuyama et al., 2012* | RRID:BDSC_66542 | |
| Genetic reagent (*D. melanogaster*) | UAS-Epac1-camps | Bloomington Drosophila Stock Center | RRID:BDSC_25407 | |
| Genetic reagent (*D. melanogaster*) | UAS-CG5955[RNAi#1] | Vienna Drosophila Resource Center | RRID: FlyBase_FBst0452036 | V15838 |
| Genetic reagent (*D. melanogaster*) | UAS-CG5955[RNAi#2] | Bloomington Drosophila Stock Center | RRID: BDSC_64566 | |
| Genetic reagent (*D. melanogaster*) | UAS-Kir | PMID: 11222642 | | |
| Genetic reagent (*D. melanogaster*) | UAS-GABA$_B$-R1[RNAi#1] | Vienna Drosophila Resource Center | RRID: FlyBase_FBst0473313 | V101440 |
| Genetic reagent (*D. melanogaster*) | UAS-GABA$_B$-R1[RNAi#2] | Vienna Drosophila Resource Center | RRID: FlyBase_FBst0490977 | V330042 |
| Genetic reagent (*D. melanogaster*) | UAS-GABA$_B$-R1[RNAi#3] | Bloomington Drosophila Stock Center | RRID:BDSC_51817 | T51817 |
| Genetic reagent (*D. melanogaster*) | UAS-GABA$_B$-R2[RNAi#1] | Vienna Drosophila Resource Center | RRID: FlyBase_FBst0452890 | V1784 |
| Genetic reagent (*D. melanogaster*) | UAS-GABA$_B$-R2[RNAi#2] | Vienna Drosophila Resource Center | RRID: FlyBase_FBst0452896 | V1785 |
| Genetic reagent (*D. melanogaster*) | UAS-GABA$_B$-R3[RNAi#1] | Vienna Drosophila Resource Center | RRID: FlyBase_FBst0468888 | V50176 |
| Genetic reagent (*D. melanogaster*) | UAS-GABA$_B$-R3[RNAi#2] | Vienna Drosophila Resource Center | RRID: FlyBase_FBst0477558 | V108036 |
| Chemical compound, drug | EOS | Tokyo Chemical Industry | Cat. No. S0445 | |
| Chemical compound, drug | NipA | Sigma | Cat. No. 211672 | |
| Chemical compound, drug | THIP | Tocris | Cat. No. 0807 | Also known as gaboxadol, 2000x stock |
| Chemical compound, drug | SKF-97541 | Tocris | Cat. No. 0379 | 10000x stock |
| Chemical compound, drug | GABA | Acros | Cat. No. AC103280250 | 10x stock |

*Continued on next page*

*Continued*

| Reagent type (species) or resource | Designation | Source or reference | Identifiers | Additional information |
|---|---|---|---|---|
| Chemical compound, drug | Pyruvate | Sigma | Cat. No. P2256 | |
| Chemical compound, drug | Tetrodotoxin (TTX) | Alomone Labs | Cat. No. T-550 | 1000x stock |
| Chemical compound, drug | caffeine | Alfa Aesar | Cat. No. A10431 | 1000x stock |
| Antibody | Mouse anti-GFP, monoclonal | UC Davis/NIH NeuroMab Facility | RRID:AB_10671955 | 1:1000 dilution |
| Antibody | Rabbit anti-GABA, polyclonal | Sigma | RRID:AB_477652 | 1:2000 dilution |
| Antibody | Rabbit anti-TH, polyclonal | Millipore | RRID:AB_390204 | 1:1000 dilution |
| Antibody | Donkey anti-Mouse AF488 | Jackson Immunoresearch | RRID:AB_2340846 | 1:600 dilution |
| Antibody | Donkey anti-Rabbit AF594 | Jackson Immunoresearch | RRID:AB_2340621 | 1:600 dilution |

## Sleep analyses

All behavioral tests were performed using individual male flies, unless otherwise indicated. Each fly was housed in a 65 × 5 mm glass tube containing 5% sucrose and 2% agar (behavior food). For amino acid supplements, the indicated amount of each amino acid was dissolved in the behavior food. For oral administration of GABA-T or GABA transporter inhibitors, 10 mM of EOS (Tokyo Chemical Industry) or 10 mg/ml of NipA (Sigma) was directly dissolved in the behavior food containing the indicated amount of threonine. For oral administration of GABA receptor agonists, 10 mg/ml of THIP (Tocris) or SKF-97541 (Tocris) stock solution was diluted into the behavior food at the indicated final concentration. Flies were fed on amino acid- and/or drug-containing behavior food in LD cycles at 25°C for 4.5 days. Locomotor activity was recorded using the DAM system (Trikinetics) and quantified by the number of infrared beam crosses per minute. Sleep bouts were defined as no activity for >5 min. Sleep parameters were analyzed using an Excel macro (*Pfeiffenberger et al., 2010*).

## Measurements of arousal threshold and sleep latency after arousal

The arousal threshold to mechanical stimuli was measured as described previously (*Wu et al., 2008*) with minor modifications. Locomotor activities were recorded similarly as in the sleep analyses, while behavioral test tubes containing individual male flies were scraped with a thin wood stick at zeitgeber (ZT) 16 (lights-on at ZT0; lights-off at ZT12) during the fourth LD cycle. Mechanical stimuli used in our tests include: 1) scraping sound and vibration without direct scraping (a weak stimulus), 2) gentle scraping (a moderate stimulus), and 3) hard scraping repeated 3–4 times (a strong stimulus). Flies were defined as aroused if they displayed inactivity for >5 min prior to the stimulus but showed any stimulus-induced locomotor response within 10 min. The percentage of aroused flies was calculated per each group in individual experiments and averaged from three independent experiments. Latency to sleep onset after the arousal was calculated in individual flies and averaged per each group. To measure the arousal threshold to a light stimulus, LD-entrained flies were exposed to a 1 min light pulse at ZT16 instead of the mechanical stimuli. The percentage of light-aroused flies and sleep latency after the light-induced arousal were measured similarly as above.

## Video analyses of sleep and locomotor activity

Wild-type male flies were placed individually into the video-tracking arena (diameter x height = 16 mm x 2 mm) in a 24-well plate filled with the behavior food (5% sucrose +2% agar±25 mM threonine) (day 0). Flies were entrained in 12 hr light:12 hr dim red light (red LED) cycles at 25°C before 24 hr time-lapse images were obtained at 0.3–1 Hz using HandyAVI software (AZcendant) on day 4. Their positional changes in X- and Y-axes were calculated from two consecutive frames of the time-lapse

images per each arena. Any positional difference larger than eight pixels was considered as a movement. A window of the time frames with no positional change for >5 min was defined as a sleep bout. Additional parameters for sleep or locomotor activity were analyzed using Excel. For the higher-resolution analysis of locomotor behaviors, male flies were pre-fed on control or amino acid-containing behavior food for four LD cycles at 25℃. After brief anesthetization, flies were individually placed into 6-well plates (diameter x height = 35 mm x 2 mm). After 25 min of habituation, time-lapse images were obtained at 10 Hz using HandyAVI software (AZcendant). Approximately 3000 frames (corresponding to a 5 min video recording) were analyzed using ImageJ software to quantify locomotor activity in individual flies as described above.

## Aversive phototaxic suppression (APS)

An APS-based short-term memory test was performed as described previously (*Dissel et al., 2015*; *Seugnet et al., 2009*). Briefly, adult male flies were individually housed and fed either control or threonine-containing behavior food for four LD cycles. A single fly was placed in the dark chamber of a T-maze without anesthesia. A filter paper (3M) was soaked with 180 μL of 1 μM quinine hydrochloride solution (Sigma) and was placed in the light chamber to give aversive condition in concordance with a light stimulus. After 1 min of habituation in the T-maze, a middle bridge between two chambers was opened and the light source was gradually turned on. Any fly which did not move to the light chamber at the first trial was excluded from further analysis. If a fly entered the light chamber within 20 secs, it was considered as a pass. The whole procedure was repeated 16 times in four sessions (four trials/session) at 1 min intervals. A performance index was calculated per each fly by the percentage of 'non-pass' in the last session.

## Whole-brain imaging

Transgenic flies were fed on control or amino acid-containing behavior food for four LD cycles at 25℃ prior to imaging experiments. Whole brains were dissected in phosphate-buffered saline (PBS) and fixed in PBS containing 3.7% formaldehyde. Fixed brains were washed three times in PBS containing 0.3% Triton X-100 (PBS-T), blocked in PBS-T containing 0.5% normal goat serum, and then incubated with mouse anti-GFP (NeuroMab) and rabbit anti-GABA (Sigma) antibodies for 2 days at 4℃. After washing three times in PBS-T, brains were further incubated with anti-mouse Alexa Fluor 488 and anti-rabbit Alexa Fluor 594 antibodies (Jackson ImmunoResearch) for 1 day at 4℃, washed three times with PBS-T, and then mounted in VECTASHIELD mounting medium (Vector Laboratories). Confocal images of whole-mount brains were acquired using a Multi-Photon Confocal Microscope (LSM780NLO, Carl Zeiss) and analyzed using ImageJ software.

## In vivo Epac1-camps imaging

Transgenic flies fed either control or threonine-containing behavior food were anesthetized in ice. A whole brain was briefly dissected in hemolymph-like HL3 solution (5 mM HEPES pH 7.2, 70 mM NaCl, 5 mM KCl, 1.5 mM $CaCl_2$, 20 mM $MgCl_2$, 10 mM $NaHCO_3$, 5 mM Trehalose, 115 mM Sucrose) and then placed on a 25 mm round coverslip. A magnetic imaging chamber (Chamlide CMB, Live Cell Instrument) was assembled on the coverslip and filled with 900 μl of HL3 solution. Where indicated, 100 μl of 10x GABA stock solution in HL3 was added to the imaging samples. Live-brain images were acquired at ~1 Hz using a multi-photon confocal microscope (LSM780NLO, Carl Zeiss) with a Plan-Apochromat 40x/1.3 oil lens. The power of a 458 nm-laser projection was 3% at a pixel resolution of $256 \times 256$. Each frame constituted two slices by ~5 μm of step sizes. Gallium arsenide phosphide (GaAsP) detectors were set by two ranges (473–491 nm and 509–535 nm) for ECFP and EYFP channels, respectively. Pinhole was fully opened (599 μm) to avoid any subtle z-drift during the image acquirement. The fluorescence intensities of CFP and YFP were quantified using ZEN software (Carl Zeiss) and any changes in FRET signals were calculated in Excel.

## Quantitative PCR

Total RNA was purified from 10 flies per each genotype (five males and five females) using Trizol Reagent, according to the manufacturer's instructions (Thermo Fisher Scientific). cDNA was prepared from DNase I-treated RNA samples using the M-MLV Reverse Transcriptase reagent (Promega) and random hexamers. Diluted cDNA samples were quantitatively analyzed by SYBR Green-

based Prime Q-Mastermix (GeNet Bio) and gene-specific primers using the LightCycler 480 real-time PCR system (Roche). To validate the efficiency of transgenic RNA interference, total RNAs from head or body extracts were analyzed similarly.

### Quantification of threonine levels

Quantitative measurement of threonine was performed as described previously (*Liu et al., 2014b*) with minor modifications. Briefly, 30 female flies were homogenized in 200 µL of PBS containing 0.05% Triton X-100. Whole-body extracts were clarified twice by centrifugation, and total proteins in the extracts were quantified using the Pierce BCA Protein Assay Kit according to manufacturer's instructions (Thermo Fisher Scientific). After boiling, soluble extracts were further clarified by centrifugation and subjected to an enzymatic reaction. Each reaction mixture included 40 µL of $5 \times$ HEPES reaction buffer (500 mM HEPES pH 8.0, 1 mM NADH, 0.25 mM pyridoxal 5-phosphate, and 5 mM dithiothreitol), 160 µL of soluble body extracts, and 1 U of alcohol dehydrogenase (Sigma). In parallel, control reactions with a serial dilution of threonine stock solution (16 mM) were used to generate a standard curve for quantification. The enzymatic reactions were set up in a 96-well microplate (Corning) and incubated for 30 min at 4°C followed by 10 min incubation at 25°C. Absorbance at 340 nm was measured for each reaction mixture using an Infinite M200 microplate reader (Tecan) before 1 µL of bacterially purified L-threonine aldolase (LTA) was added to each reaction mixture. The reaction mixture was further incubated at 37°C for 5 min and post-LTA absorbance was measured to calculate decreases in NADH levels.

### Protein purification of L-threonine aldolase

The coding sequence of LTA was PCR-amplified from genomic DNA of *Pseudomonas aeruginosa* (a gift from R.J. Mitchell) and cloned into a modified pDuet vector (a gift from C. Lee). Bacterial purification of His-tagged LTA proteins using Ni-NTA Agarose (Qiagen) was performed as described previously (*Lee et al., 2017*). Purified proteins were dialyzed using a dialysis buffer (50 mM $NaH_2PO_4$, pH 8.0, 10 µM pyridoxal 5-phosphate, and 1 mM dithiothreitol), diluted in 50% glycerol, quantified using Pierce BCA Protein Assay Kit (Thermo Fisher Scientific), and stored at −80°C prior to use.

### Quantitative analyses of free amino acids and energy metabolites

Wild-type male flies were loaded on to standard cornmeal-yeast-agar food containing either 0 mM (control) or 50 mM threonine, and then entrained in LD cycles at 25°C for 4 days before harvest. Extracts were prepared from 100 fly heads per condition and the relative levels of free amino acids were measured using ion exchange chromatography as described previously (*Sonn et al., 2018*). For quantification of energy metabolites, fly heads were homogenized in 400 µl of chloroform/methanol (2/1, v/v) and clarified by centrifugation. The supernatant was dried by vacuum centrifugation, and then reconstituted with 50 µL of 50% acetonitrile prior to liquid chromatography-tandem mass spectrometry analysis using 1290 HPLC (Agilent), Qtrap 5500 (ABSciex), and a reverse phase column (Synergi fusion RP $50 \times 2$ mm).

### Statistics

Appropriate sample sizes were not determined by statistical computation but based on those reported in previous studies. For all the analyses, 'n' refers to the total number of biological replicates which were tested in more than two independent experiments, unless otherwise indicated in figure legends. For immunofluorescence assay, 'n' refers to the total number of brain hemispheres which were tested in 2–4 independent experiments. For cAMP imaging, 'n' refers to the total number of brains which were tested in 2–3 independent experiments. Individual flies were allocated into each group of biological replicates by their specific diet or genotype. Raw sleep data were collected non-blindly to the conditions but analyzed by an automated macro program. For immunofluorescence assay, GFP-positive neurons were scored in a way of double-blinded to the conditions. Short-term memory tests were performed blindly to the conditions. All the statistical analyses were performed using Prism (GraphPad Software, Inc) as described in figure legends. F distributions with degrees of freedom were indicated by F[DFn, DFd]. All the *P* values from post hoc tests after one-way or two-way ANOVA were corrected for multiple comparisons. Violin plots present mean ± 95%

confidence intervals and were generated using Python with the help of Seaborn library. Bar graphs indicate mean ± SEM and were generated using Excel.

## Acknowledgements

We thank JM Han at Yonsei University College of Pharmacy for conceiving relevant ideas at the initial stage of the research; JY Sonn at KAIST for critical reading of the manuscript and helpful comments; C Lee and RJ Mitchell at UNIST for reagents; EY Suh at Chungnam National University for amino acid analyses; SJ Kim and HJ Yoo at University of Ulsan College of Medicine for energy metabolite analyses; A Sehgal at University of Pennsylvania School of Medicine, Bloomington *Drosophila* stock center, Korea *Drosophila* resource center, Kyoto stock center, and Vienna *Drosophila* resource center for *Drosophila* strains.

## Additional information

### Funding

| Funder | Grant reference number | Author |
| --- | --- | --- |
| National Research Foundation of Korea | NRF-2017R1E1A2A02066965 | Chunghun Lim |
| National Research Foundation of Korea | NRF-2018R1A5A1024261 | Chunghun Lim |
| Suh Kyungbae Foundation | SUHF-17020101 | Chunghun Lim |

The funders had no role in study design, data collection and interpretation, or the decision to submit the work for publication.

### Author contributions

Yoonhee Ki, Software, Validation, Investigation, Visualization, Methodology, Writing—original draft; Chunghun Lim, Conceptualization, Supervision, Funding acquisition, Validation, Investigation, Writing—original draft, Project administration, Writing—review and editing

### Author ORCIDs

Chunghun Lim (iD) https://orcid.org/0000-0001-8473-9272

### Decision letter and Author response

Decision letter https://doi.org/10.7554/eLife.40593.031
Author response https://doi.org/10.7554/eLife.40593.032

## Additional files

### Supplementary files

• Transparent reporting form
DOI: https://doi.org/10.7554/eLife.40593.029

### Data availability

All data generated or analysed during this study are included in the manuscript and supporting files.

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
