## [Decision Letter]

[**Editorial note:** This article has been through an editorial process in which the authors decide how to respond to the issues raised during peer review. The Reviewing Editor's assessment is that all the issues have been addressed.]

Thank you for submitting your article "Threonine enhances sleep drive via a circadian clock-independent GABAergic pathway in for consideration by *eLife*. Your article has been reviewed by two peer reviewers, and the evaluation has been overseen by K VijayRaghavan as the Reviewing Editor and Senior Editor. The reviewers have opted to remain anonymous.

The Reviewing Editor has highlighted the concerns that require revision and/or responses, and we have included the separate reviews below for your consideration. If you have any questions, please do not hesitate to contact us.

Summary:

This manuscript from the Lim laboratory follows upon their recent work showing that serine modulates sleep in *Drosophila*. The focus here is on threonine, which appears to be sleep-promoting, likely through an effect on GABA signaling. The idea that metabolic factors are important regulators of sleep is gaining recognition, and this study is an interesting addition in that regard. As written though, this big picture view of the work does not come across until the end of the Discussion; laying it out earlier in the manuscript (e.g. Abstract and Introduction) would significantly boost the appeal of the work. From a scientific perspective, it would be good to have the mechanism nailed. This will require attention to the major points in each of the reviews below. Attending to these will likely involve experiments that will take about 2-3 months in addition to the time needed for re-writing parts of the manuscript.

Major concerns that require revision and/or responses.

These concerns derive from our consultation on the two reviews. They highlight specific points and need to be seen along with the two reviews.

1) The scope of the paper's title gets lost in detail. As pointed out in the summary above, the big overview needs to be linked to the core of every section of the Results.

2) The fundamental matter is that the authors have not nailed the mechanism. Ideally, they should show how GABA signaling is affected, but at the very least address the points made by the reviewers or satisfactorily exclude clock cells

3) The authors suggest that Threonine and GABA are competitive substrates. Yet, the results suggest that an explanation could be at the level of GABA synthesis presynaptically (Figure 4—figure supplement 6). This need to be better discussed.

4) Both reviewers have pointed out the need to connect neuronal CalExa signals with threonine to sleep in a robust way. For example, do GAD+ cells with increased CaLexA signal have anything to do with sleep? Although this may be much more difficult to show. (Are there known GAD+, sleep-relevant drivers the authors could use?)

5) The GAD-Gal4 shibire experiment is not making a persuasive point. Maybe it should be repeated? Please particularly note the points about rutabaga made by reviewer #1. Both reviewers feel that this is an important experimental point to address.

6) With regard to reviewer #2's point 3, our consultation agreed in the general principle that multiple alleles should be used for each gene, but if some of these genes fall into the same pathway and/or corroborate findings made in other ways (pharmacology), then this could be seen as mitigating,

To deal with potential developmental roles, it will be good to see at least one adult- specific manipulation and phenotype of GABA manipulation.

Separate reviews (please respond to each point):

*Reviewer #1:*

The authors favor the idea that threonine acts through GABA signaling, pointing out that threonine and GABA and their metabolic derivatives are competitive substrates? For what-for the GABA receptor? This would explain why GABA-T mutants are unresponsive to threonine as they have high extracellular GABA, and would be consistent with the modest block produced by threonine in GABA action in an ex vivo preparation. However, how does this explain interactions of threonine with presynaptic GABA-CaLexa signals are increased in GAD neurons by threonine and blocking GAD neurons with shibire is stated to attenuate effects of threonine?

Does the CaLexa signal increase in non-GABA cells with threonine treatment? The manuscript shows specificity for threonine by using another amino acid as control (arginine), but what about specificity for GABAergic cells?

The blocking effect of GAD-GAL4 driven shibire on threonine action is questionable (Figure 4) as the threonine effect in controls in this experiment is even smaller than normal, and even normally it is not great. Is this really relevant (even if significant)?

The comparisons in Figure 4B are hard to follow. If the point is that threonine has no effects when GABA_B_-R1 is knocked down, then it should be shown that threonine increases sleep in the GAL4 alone control, but not in the knockdown, relative to the same genotypes not treated with threonine (from the data, it looks like threonine does increase sleep, probably significantly, in the knockout). Instead, control and knockdown are compared with each other, both in the presence of threonine. What this says is that knockdown of GABA_B_-R1 does not have any effect in threonine-treated flies, which is not the point (I think).

If the learning assay with rutabaga is to show that increased sleep (with threonine) rescues learning in rutabaga, then the authors should show that blocking the sleep increase (with deprivation) abrogates rescue. Otherwise, the direct effects of threonine on learning cannot be excluded.

Does threonine or the threonine-increasing mutant display any kind of metabolic phenotype, as does the GABAT mutant (Maguire et al., 2015)? Given that the underlying mechanisms are the same or overlapping, it is worth at least discussing this point.

Additional data files and statistical comments:

The authors use Student's t-test a lot, which is not ideal or stringent enough for most measurements.

*Reviewer #2:*

The manuscript by Ki and Lim demonstrates sleep-promoting effects of dietary threonine (SPET) in *Drosophila*. The authors make several interesting claims regarding the mechanism of SPET: 1) decreased sleep latency in response to dietary threonine (Thr) does not require functional clock genes or pacemaker neurons; 2) GABA signaling through GABA_B_-R1 is involved in SPET; and 3) Thr activates a subset of GABAergic neurons, reduces GABA responsiveness of sleep-relevant R2 EB neurons, and rescues memory defects of rutabaga mutants; and 4) genetic manipulations that increase endogenous Thr levels enhance sleep. These are potentially interesting findings, but additional data are required to strengthen some of the claims.

Major points:

1) To demonstrate the relevance to GABA signaling for SPET, the authors show increased CaLexA signal in response to Thr in neurons lateral to the antennal lobe, which they call LNs. However, they do not show that these neurons have any role in sleep. Nor do they show any increase in CaLexA signal in known sleep centers in response to Thr.

2) The authors claim that a slightly reduced GABA response to Thr in R2 EB neurons implies that threonine-induced behavioral quiescence is physiologically relevant to sleep. However, it is unclear whether the effect is sufficient to contribute significantly to SPET. Does R2 EB specific knockdown of GABA_B_-R1 affect SPET?

3) The involvement of GABA-T in SPET is supported by a single allelic combination (hypomorph over deficiency) and that of GABA_B_-R1 is based on a single RNAi construct. The involvement of *CG5955* is based on one allelic combination and one RNAi. To show that a gene is involved in a process, it is important to provide converging evidence from multiple sources: e.g., multiple mutant alleles or allelic combinations, multiple RNAi constructs to minimize the potential off-target effects, and rescue with transgene expression.

4) Potential developmental defects could be a confounding factor for the effects of GABA-T, GABA_B_-R1, and *CG5955* on SPET. Adult specific knockdown or developmental rescue could alleviate the concern.

5) The data behind the claim that *Rdl* and PDF neurons are not involved in SPET are not compelling. In Figure 2—figure supplement 1B, *Rdl* MMDR/1 mutants seem to have reduced SPET (i.e., the difference between – and + Thr is smaller in mutants than het controls). If the reduction is not statistically significant, it may be due to the unusually large error bar in one of the conditions. Additional data may provide a clearer picture.

A related concern is that without Thr, sleep latency is not reduced in *Rdl* MMDR/1 mutants relative heterozygous controls (B), as would be expected based on previous reports. Similarly, without Thr, PDF and CRY neuron silencing does not lead to sleep latency reduction (C). These unexpected features of the data raise concerns about the validity of the rest of the data.

6) Rescue of memory defects in rut mutants by Thr is interesting. However, it would strengthen the conclusion if additional memory mutants were tested. Please see reviewer 1 's points on this,

Minor Comments:

1) It is unclear how the effects of Thr vary as a function of time-of-day. It would be informative to show these data in a format where sleep amount per 30min or 1h is plotted as a function of time of day.

2) To further examine the clock- and light-dependence of SPET, it would be helpful to repeat the basic +/- Thr experiments in DD and LL.

3) As the authors note, GABA-T effects on SPET could be due to a floor effect. Although pharmacological data somewhat alleviate this concern, a genetic demonstration (using hypomorphs?) would strengthen the claim.

4) In Figure 1A and related figures, it is unclear whether p values were corrected for multiple comparisons.

5) Figure 1—figure supplement 3. In this and other similar figures, to determine whether SPET is affected by an experimental manipulation (in this case, silencing of gustatory or olfactory neurons), it is important to test whether the difference between + and – Thr is different across the three genotypes.

6) Discussion section: "…SPET facilitates sleep onset in a manner independent of light". Where are the data for light-independence of SPET?

7) Figure 2—figure supplement 1A and B. Some of the error bars are so big that it is difficult to believe the differences are significant at p<0.001 as indicated. In these situations, standard ANOVAs that assume equal variance across conditions are not appropriate.

8) In Figure 3A, the CaLexA signal for LNs is not symmetric. Is this common?

9) Figure 4B is formatted differently from all other figures. Why?

10). Some of the N's for behavioral experiments are as low as 8. This seems too low.

11) SEM, which takes into account N's, would more informative than SD.

Additional data files and statistical comments:

As noted in the minor point #7, some of the statistics do not seem appropriate.

---

## [Author Response]

Major concerns that require revision and/or responses.These concerns derive from our consultation on the two reviews. They highlight specific points and need to be seen along with the two reviews.1) The scope of the paper's title gets lost in detail. As pointed out in the summary above, the big overview needs to be linked to the core of every section of the Results.

We modified our title, Abstract, and Introduction to better state our view regarding the metabolic control of sleep behaviors in the revised manuscript.

2) The fundamental matter is that the authors have not nailed the mechanism. Ideally, they should show how GABA signaling is affected, but at the very least address the points made by the reviewers or satisfactorily exclude clock cells

In our revised manuscript, we provided new pieces of evidence that exclude the possible requirement of clock-dependent control of sleep in SPET. First, SPET was detectable even in constant light (LL) where wild-type flies lose circadian rhythms and sleep-wake cycles (Figure 2C and D; Figure 2—figure supplement 2). Second, a genetic loss of *per* or *Clk* shortened or lengthened sleep latency, respectively, in LD cycles (Liu et al., 2014) while these arrhythmic clock mutants displayed SPET robustly (Figure 2A). Third, PDF neuron-specific overexpression of dominant-negative CLK proteins lengthened sleep latency (Liu et al., 2014) whereas the loss of molecular clocks in PDF neurons did not compromise SPET (Figure 2B).

We further showed that transgenic depletion of metabotropic GABA receptors in R2 EB neurons promoted sleep non-additively with SPET (Figure 5C). Given that dietary threonine reduced metabotropic GABA responses in R2 EB neurons, these results support that SPET may involve the down-regulation of metabotropic GABA transmission in this specific neural locus for generating homeostatic sleep drive. Please see our point-by-point responses to additional points made by each reviewer below.

3) The authors suggest that Threonine and GABA are competitive substrates. Yet, the results suggest that an explanation could be at the level of GABA synthesis presynaptically (Figure 4—figure supplement 6). This need to be better discussed.

We actually found that GABA levels were low in threonine-fed flies compared to control (Figure 4A). While this could be a compensatory decrease, additional lines of our evidence indicated that suppression of the metabotropic GABA transmission indeed induced SPET-like sleep phenotypes in control-fed flies and displayed non-additive effects with SPET in threonine-fed flies (please see our responses to the reviewer comment #1-1 and the reviewer #2, major point #2 below). We thus reasoned that dietary threonine decreased GABA levels likely as a metabolic consequence important for SPET and better discussed it in our revised manuscript.

4) Both reviewers have pointed out the need to connect neuronal CalExa signals with threonine to sleep in a robust way. For example, do GAD+ cells with increased CaLexA signal have anything to do with sleep? Although this may be much more difficult to show. (Are there known GAD+, sleep-relevant drivers the authors could use?)

Given our revised model for the implication of GABA transmission in SPET, we toned down our conclusion from the CaLexA experiment and discussed a possible explanation for the CaLexA phenotypes in the revised manuscript. Please see our specific responses to the reviewer comments #1-1 and #1-2 as well as the reviewer #2, major points #1 and 2 below.

5) The GAD-Gal4 shibire experiment is not making a persuasive point. Maybe it should be repeated?

In our revised manuscript, we provided more convincing data from the GAD1-Gal4 shibire experiments (Figure 4B). Please see our response to the reviewer comment #1-3 below.

Please particularly note the points about rutabaga made by reviewer #1. Both reviewers feel that this is an important experimental point to address.

In our revised manuscript, we provided new data that dietary threonine rescued memory deficits in another memory mutants (Figure 6B). Using caffeine-induced sleep suppression, we further showed that this memory rescue required threonine-induced sleep (Figure 6B and C). Please see our responses to the reviewer comment #1-5 and the reviewer #2, major point #6.

6). With regard to reviewer #2's point 3, our consultation agreed in the general principle that multiple alleles should be used for each gene, but if some of these genes fall into the same pathway and/or corroborate findings made in other ways (pharmacology), then this could be seen as mitigating,To deal with potential developmental roles, it will be good to see at least one adult- specific manipulation and phenotype of GABA manipulation.

In our revised manuscript, we provided converging evidence from multiple genetic and transgenic sources (e.g., multiple allelic combinations, transgenic rescue, multiple RNAi transgenes) to validate the function of a given gene and the implication of a metabotropic GABA pathway in threonine-relevant sleep regulation. While our pharmacological manipulations of GABA transmission independently confirmed the genetic evidence, oral administration of GABA-relevant inhibitors and agonists to adult flies also excluded the possible developmental effects of GABA manipulations. These adult-specific, pharmacological manipulations of GABA transmission included 1) GABA-T inhibitor (Figure 3B), 2) GABA transporter inhibitor (Figure 3B), and 3) agonists of ionotropic or metabotropic GABA receptors (Figure 4D). We further showed that a conditional blockade of synaptic transmission from adult neurons expressing GAD1 impaired SPET, supporting adult-specific effects of the transgenic GABA manipulation on SPET (Figure 4B). Please see our specific responses to reviewer #2, major points #3 and #4 below.

Separate reviews (please respond to each point):

Reviewer #1:

*The authors favor the idea that threonine acts through GABA signaling, pointing out that threonine and GABA and their metabolic derivatives are competitive substrates? For what-for the GABA receptor? This would explain why GABA-T mutants are unresponsive to threonine as they have high extracellular GABA, and would be consistent with the modest block produced by threonine in GABA action in an* ex vivo *preparation. However, how does this explain interactions of threonine with presynaptic GABA-CaLexa signals are increased in GAD neurons by threonine and blocking GAD neurons with shibire is stated to attenuate effects of threonine?*

The structural homology among threonine, GABA, and their derivatives led us to the hypothesis that these relevant chemicals may act as competitive substrates in enzymatic reactions for their overlapping metabolism. As the reviewer suggested above, they may similarly act as competitive ligands for GABA receptors (i.e., metabotropic GABA receptors, in particular), explaining weaker GABA responses in R2 EB neurons of threonine-fed flies than those in control.

In our revised manuscript, we further found that threonine-fed flies displayed low levels of GABA and glutamate compared to those in control-fed flies (Figure 4A). Although this could be a compensatory decrease, a simple hypothesis would be that dietary threonine limits the total flux of GABA/glutamate/glutamine cycle possibly through substrate competition, decreases the size of available GABA pool, and thereby down-scales GABA transmission for SPET. This model is consistent with our observations that genetic suppression of GABA transmission by the conditional blockade of synaptic transmission in GAD1-expressing neurons or by the transgenic depletion of metabotropic GABA receptors in R2 EB neurons drives sleep in control-fed flies (i.e., mimics SPET in control-fed flies) and displays non-additive effects with SPET in threonine-fed flies. By contrast, high levels of extracellular GABA (e.g., *GABA-T* mutants or EOS/NipA-fed flies) or pharmacological enhancement of the metabotropic GABA transmission (e.g., oral administration of the GABA-B receptor agonist) may interfere with this process, thereby conferring the resistance to SPET.

Given this revised model, we reason that some group of GAD1 neurons (i.e., LN) may express presynaptic GABA receptors to suppress their own GABA transmission via a negative feedback. Threonine diet could then de-repress the auto-inhibitory GABA transmission, elevating the CaLexA signals selectively in those GAD1 neurons. Accordingly, we revised our Results and Discussion to clarify this model and better interpret our results.

Does the CaLexa signal increase in non-GABA cells with threonine treatment? The manuscript shows specificity for threonine by using another amino acid as control (arginine), but what about specificity for GABAergic cells?

As the reviewer suggested, we tested additional Gal4 drivers expressed in other sleep-regulatory loci (i.e., mushroom body and dopaminergic neurons) in the CaLexA experiments and did not see any detectable increase in their CaLexA signals upon threonine diet (Figure 4—figure supplement 4D and E).

The blocking effect of GAD-GAL4 driven shibire on threonine action is questionable (Figure 4) as the threonine effect in controls in this experiment is even smaller than normal, and even normally it is not great. Is this really relevant (even if significant)?

As the editor suggested above, we repeated the shibire experiment in a different incubator with better control of the internal temperature and provided more convincing data in our revised manuscript (Figure 4B). We confirmed that overexpression of the temperature-sensitive shibire in GAD1-expressing neurons induced sleep in control-fed condition and masked SPET only at restrictive temperature. We also noted that low (i.e., permissive) temperature delayed SPET even in heterozygous controls, explaining smaller SPET in our original data.

The comparisons in Figure 4B are hard to follow. If the point is that threonine has no effects when GABA_B_-R1 is knocked down, then it should be shown that threonine increases sleep in the GAL4 alone control, but not in the knockdown, relative to the same genotypes not treated with threonine (from the data, it looks like threonine does increase sleep, probably significantly, in the knockout). Instead, control and knockdown are compared with each other, both in the presence of threonine. What this says is that knockdown of GABA_B_-R1 does not have any effect in threonine-treated flies, which is not the point (I think).

We thought that the comparisons in the original Figure 4B would better visualize the effects of neuronal GABA_B_-R1 depletion on SPET among other genotypes. Given the comments from both reviewers (please also see the reviewer #2, minor comment #9), we realized that the different format was rather confusing. Accordingly, we presented all our behavioral data in the same format in the revised manuscript.

If the learning assay with rutabaga is to show that increased sleep (with threonine) rescues learning in rutabaga, then the authors should show that blocking the sleep increase (with deprivation) abrogates rescue. Otherwise, the direct effects of threonine on learning cannot be excluded.

In our revised manuscript, we showed that dietary threonine also rescued memory deficits in *dumb* mutants (Figure 6B). Co-administration of caffeine with threonine substantially blocked sleep induction as well as memory rescue in *dumb* mutants (Figure 6B and C). By contrast, caffeine alone did not significantly affect short-term memory in control and *dumb* mutants under our experimental condition. These data thus support that dietary threonine rescues memory mutants in a sleep-dependent manner.

Does threonine or the threonine-increasing mutant display any kind of metabolic phenotype, as does the GABAT mutant (Maguire et al., 2015)? Given that the underlying mechanisms are the same or overlapping, it is worth at least discussing this point.

To examine any metabolic phenotypes induced by dietary threonine, we compared the relative levels of free amino acids and energy metabolites between control- and threonine-fed flies. Threonine diet lowered GABA and glutamate levels, but it did not significantly affect ATP levels or the ratio of NAD+ to NADH (Figure 4A and Figure 4—figure supplement 1). While dietary glutamate rescued metabolic phenotypes in *GABA-T* mutants (Maguire et al., 2015), it affected neither long sleep in *GABA-T* mutants (Maguire et al., 2015) nor SPET (Figure 4—figure supplement 3), indicating that low glutamate levels do not explain either sleep phenotypes. Accordingly, we compared and discussed these metabolic phenotypes in *GABA-T* mutants and threonine-fed flies in our revised manuscript.

Additional data files and statistical comments:The authors use Student's t-test a lot, which is not ideal or stringent enough for most measurements.

Wherever possible, we avoided Student’s t-test to increase the stringency for our statistical analyses in the revised manuscript. We also clarified our statistical analyses in the text, figures, and figure legends of our revised manuscript.

Reviewer #2:

The manuscript by Ki and Lim demonstrates sleep-promoting effects of dietary threonine (SPET) in Drosophila. The authors make several interesting claims regarding the mechanism of SPET: 1) decreased sleep latency in response to dietary threonine (Thr) does not require functional clock genes or pacemaker neurons; 2) GABA signaling through GABA_B_-R1 is involved in SPET; and 3) Thr activates a subset of GABAergic neurons, reduces GABA responsiveness of sleep-relevant R2 EB neurons, and rescues memory defects of rutabaga mutants; and 4) genetic manipulations that increase endogenous Thr levels enhance sleep. These are potentially interesting findings, but additional data are required to strengthen some of the claims.Major points:1) To demonstrate the relevance to GABA signaling for SPET, the authors show increased CaLexA signal in response to Thr in neurons lateral to the antennal lobe, which they call LNs. However, they do not show that these neurons have any role in sleep. Nor do they show any increase in CaLexA signal in known sleep centers in response to Thr.

As mentioned in our response to the reviewer comment #1-1 above, we did not see any detectable increase in the CaLexA signals from sleep-regulatory mushroom body or dopaminergic neurons upon threonine diet (Figure 4—figure supplement 4D and E). Although the sensitivity of CaLexA may intrinsically limit the detectable size and duration of Ca^2+^ changes in our experiment, these results support the relative specificity of Ca^2+^ response in LN to the threonine diet. Lack of a specific Gal4 driver that targets these LN only, however, did not allow us to validate that these effects were necessary for SPET. Nonetheless, it led us to find that a conditional blockade of synaptic transmission in GAD1-expressing neurons, including LN, actually induced sleep in control-fed flies, and masked SPET in threonine-fed flies. Given our revised model for the implication of GABA transmission in SPET, we toned down our conclusion from the CaLexA experiment and discussed a possible explanation for the CaLexA phenotypes in the revised manuscript.

2) The authors claim that a slightly reduced GABA response to Thr in R2 EB neurons implies that threonine-induced behavioral quiescence is physiologically relevant to sleep. However, it is unclear whether the effect is sufficient to contribute significantly to SPET. Does R2 EB specific knockdown of GABA_B_-R1 affect SPET?

In our revised manuscript, we tested several RNAi transgenes to individually deplete metabotropic GABA receptors in R2 EB neurons and examined their effects on SPET. Depletion of GABAB-R2 or GABAB-R3 in R2 EB neurons indeed enhanced sleep drive non-additively with SPET (Figure 5C). This contrasted with our observations that genetic elevation of GABA levels or pharmacological enhancement of the metabotropic GABA transmission suppressed SPET, but it did not comparably affect sleep drive in control-fed flies (Figure 3A and Figure 4D). Considering our additional finding that dietary threonine reduces GABA levels, we revised our model for SPET and proposed that 1) sleep drive by low metabotropic GABA transmission in R2 EB neurons may mediate SPET, and 2) genetic/pharmacological elevation of the GABA transmission may interfere with this process, thereby masking SPET. We revised our manuscript accordingly.

3) The involvement of GABA-T in SPET is supported by a single allelic combination (hypomorph over deficiency) and that of GABA_B_-R1 is based on a single RNAi construct. The involvement of CG5955 is based on one allelic combination and one RNAi. To show that a gene is involved in a process, it is important to provide converging evidence from multiple sources: e.g., multiple mutant alleles or allelic combinations, multiple RNAi constructs to minimize the potential off-target effects, and rescue with transgene expression.

As the reviewer suggested, we provided converging evidence from multiple sources in the revised manuscript. These included 1) multiple allelic combinations (*GABA-T, CG5955*), 2) transgenic rescue (*GABA-T*), 3) multiple RNAi transgenes (metabotropic GABA receptors, *CG5955*), and 4) adult-specific, pharmacological manipulations of target gene function (*GABA-T*, GABA transporter, GABA receptors) to exclude any development effects on SPET.

4) Potential developmental defects could be a confounding factor for the effects of GABA-T, GABA_B_-R1, and CG5955 on SPET. Adult specific knockdown or developmental rescue could alleviate the concern.

We have been testing adult-specific knockdown or developmental rescue using tubGal80ts. Unfortunately, heterozygous controls of the key UAS transgenes (e.g., UAS-GABA-T, UAS-GABA receptor RNAi) displayed Gal4-independent effects on the genetic rescue or SPET at high temperature. Although it was likely due to their leaky expression, this limited our experimental conditions to genetically validate the developmental defects. Nonetheless, we provided a series of compelling evidence that supports adult-specific effects of GABA manipulations on SPET. First, adult-specific administration of GABA-T inhibitor or GABA transporter inhibitor potently suppressed SPET (Figure 3B). Second, an adult-specific blockade of synaptic transmission in GAD1-expressing cells induced sleep drive non-additively with SPET (Figure 4B). Third, adult-specific administration of the agonist of metabotropic GABA receptors, but not of ionotropic GABA receptors, suppressed SPET (Figure 4D). Regarding CG5955, we could not fully exclude the possibility of metabolic compensation or developmental effects using transgenic reagents available to us. So, we toned down our conclusion and stated these possibilities in our revised manuscript.

5) The data behind the claim that Rdl and PDF neurons are not involved in SPET are not compelling. In Figure 2—figure supplement 1B, Rdl MMDR/1 mutants seem to have reduced SPET (i.e., the difference between – and + Thr is smaller in mutants than het controls). If the reduction is not statistically significant, it may be due to the unusually large error bar in one of the conditions. Additional data may provide a clearer picture.A related concern is that without Thr, sleep latency is not reduced in Rdl MMDR/1 mutants relative heterozygous controls (B), as would be expected based on previous reports. Similarly, without Thr, PDF and CRY neuron silencing does not lead to sleep latency reduction (C). These unexpected features of the data raise concerns about the validity of the rest of the data.

In our original manuscript, we examined sleep behaviors in male trans-heterozygous mutants of *Rdl* and could not observe their short sleep latency compared to heterozygous controls in control-fed condition. Since the original paper has shown the latency phenotype in female *Rdl* mutants (Agosto et al., 2008), we examined baseline sleep as well as SPET in female flies. Consistent with the previous finding, female *Rdl* MMDR/1 mutants displayed shorter sleep latency than their heterozygous controls in control-fed condition (Figure 2—figure supplement 1). We further observed comparable SPET between *Rdl* mutants and their heterozygous controls (*P*=0.1381 for sleep amount; *P*=0.2881 for sleep latency by two-way ANOVA). These data indicate that loss of *Rdl* function does not impair SPET.

Regarding the transgenic manipulations of circadian pacemaker neurons, we tested male transgenic flies in our original manuscript and could not detect any phenotype in their sleep latency. As the reviewer mentioned above, these data were not consistent with previous observations in female flies that have suggested the wake-promoting role of PDF-expressing neurons (Parisky et al., 2008; Liu et al., 2014). Accordingly, we examined SPET in female transgenic flies that have been validated for clock-dependent control of sleep latency in PDF neurons. Consistent with the published result (Liu et al., 2014), we found that overexpression of dominant-negative CLOCK proteins only in PDF neurons was sufficient to lengthen sleep latency in female flies fed control food (Figure 2B). The loss of molecular clocks in PDF neurons did not compromise SPET on the sleep amount as well as sleep latency. These new pieces of our genetic evidence more convincingly demonstrate that SPET requires neither *Rdl*- nor PDF clock-dependent control of sleep drive.

6) Rescue of memory defects in rut mutants by Thr is interesting. However, it would strengthen the conclusion if additional memory mutants were tested. Please see reviewer 1 's points on this,

In our revised manuscript, we further showed that dietary threonine rescued memory deficits in *dumb* mutants likely in a manner dependent on threonine-induced sleep (Figure 6B). Please see our response to the reviewer comment #1-5 above.

Minor Comments:1) It is unclear how the effects of Thr vary as a function of time-of-day. It would be informative to show these data in a format where sleep amount per 30min or 1h is plotted as a function of time of day.

We included the sleep profiles of threonine-fed flies as well as their sleep latency at different time-points of the day in the revised manuscript (Figure 2C and D, Figure 2—figure supplement 2).

2) To further examine the clock- and light-dependence of SPET, it would be helpful to repeat the basic +/- Thr experiments in DD and LL.

As suggested by the reviewer, we examined SPET in LL or DD condition, and included these new data in the revised manuscript (Figure 2C and D, Figure 2—figure supplement 2).

3) As the authors note, GABA-T effects on SPET could be due to a floor effect. Although pharmacological data somewhat alleviate this concern, a genetic demonstration (using hypomorphs?) would strengthen the claim.

In our revised manuscript, we tested several mutants trans-heterozygous for *GABA-T* alleles (Figure 3—figure supplement 1A) and found that those harboring weaker allelic combinations did not display a floor effect on sleep latency, but their sleep behaviors were actually resistant to SPET (Figure 3A).

4) In Figure 1A and related figures, it is unclear whether p values were corrected for multiple comparisons.

All the *P* values from post hoc tests after one-way or two-way ANOVA were corrected for multiple comparisons. To clarify this, we included this statement in Materials and methods (Statistics section) of the revised manuscript.

5) Figure 1—figure supplement 3. In this and other similar figures, to determine whether SPET is affected by an experimental manipulation (in this case, silencing of gustatory or olfactory neurons), it is important to test whether the difference between + and – Thr is different across the three genotypes.

To determine whether or not our experimental manipulations significantly affected SPET, we compared SPET among all necessary heterozygous controls (e.g., Gal4/+, UAS/+) and testing genotypes (e.g., Gal4/UAS) by two-way ANOVA. To clarify this, we modified our text, figures, and figure legends in the revised manuscript to better describe our statistical comparisons among different genotypes and conditions.

6) Discussion section: "…SPET facilitates sleep onset in a manner independent of light". Where are the data for light-independence of SPET?

In our original manuscript, we generally measured the latency to sleep onset right after lights-off in LD cycles (ZT12) but we also observed SPET on the sleep latency after mechanical arousal at ZT16 (i.e., 4 hours after lights-off in LD cycles). These results indicate that SPET does not require the presence of light transition (Figure 1B). In our revised manuscript, we showed that SPET was also detectable in LL or DD (Figure 2C and D, Figure 2—figure supplement 2). Nonetheless, we thought that the wording, “light-independence”, might unnecessarily underestimate possible effects of light on the scaling of SPET. So, we modified our text in the revised manuscript.

7) Figure 2—figure supplement 1A and B. Some of the error bars are so big that it is difficult to believe the differences are significant at p<0.001 as indicated. In these situations, standard ANOVAs that assume equal variance across conditions are not appropriate.

Error bars in our original violin plots indicate SD although large variations in some genotypes (e.g., *Clk*[Jrk] or *Pdf*>CLK[DN]) are likely their phenotype. We revised all our violin plots to show 95% confidence intervals (corresponding to 1.98 fold of SEM).

8) In Figure 3A, the CaLexA signal for LNs is not symmetric. Is this common?

We quantified the CaLexA signals from each hemisphere since we often observed asymmetric ones in the whole-brain images. The CaLexA reporter involves a transcriptional amplification step. So, we reason that subtle differences in the levels and/or duration of Ca^2+^ influx between the same groups of neurons in each hemisphere may lead to asymmetric CaLexA signals at detectable levels.

9) Figure 4B is formatted differently from all other figures. Why?

We thought that the comparisons in the original Figure 4B would better visualize the effects of neuronal GABA_B_-R1 depletion on SPET among other genotypes. However, the reviewer #1 also mentioned that the original Figure 4B was rather hard to follow (please see the reviewer comment #1-4). Accordingly, we presented all our behavioral data in the same format in the revised manuscript.

10). Some of the N's for behavioral experiments are as low as 8. This seems too low.

We increased N in the relevant experiments.

11) SEM, which takes into account N's, would more informative than SD.

In our revised manuscript, we showed 95% confidence intervals in all the violin plots instead of SD.

Additional data files and statistical comments:As noted in the minor point #7, some of the statistics do not seem appropriate.

Please see our response to the reviewer comment #1-7 above.